# Xenogeneic silencing strategies in bacteria are dictated by RNA polymerase promiscuity

David Forrest [1], Emily A. Warman[1], Amanda M. Erkelens [2], Remus T. Dame [2,3] & David C. Grainger [1✉]

Horizontal gene transfer facilitates dissemination of favourable traits among bacteria. However, foreign DNA can also reduce host fitness: incoming sequences with a higher AT content than the host genome can misdirect transcription. Xenogeneic silencing proteins counteract this by modulating RNA polymerase binding. In this work, we compare xenogeneic silencing strategies of two distantly related model organisms: *Escherichia coli* and *Bacillus subtilis*. In *E. coli*, silencing is mediated by the H-NS protein that binds extensively across horizontally acquired genes. This prevents spurious non-coding transcription, mostly intragenic in origin. By contrast, binding of the *B. subtilis* Rok protein is more targeted and mostly silences expression of functional mRNAs. The difference reflects contrasting transcriptional promiscuity in *E. coli* and *B. subtilis*, largely attributable to housekeeping RNA polymerase σ factors. Thus, whilst RNA polymerase specificity is key to the xenogeneic silencing strategy of *B. subtilis*, transcriptional promiscuity must be overcome to silence horizontally acquired DNA in *E. coli*.

[1] School of Biosciences, University of Birmingham, Edgbaston B15 2TT, UK. [2] Leiden Institute of Chemistry, Leiden University, Einsteinweg 55, 2333CC Leiden, The Netherlands. [3] Centre for Microbial Cell Biology, Leiden University, Einsteinweg 55, 2333CC Leiden, The Netherlands. ✉email: d.grainger@bham.ac.uk

Accurate transcription initiation by RNA polymerase depends on DNA sequences called promoters[1]. In bacteria, RNA polymerase exists in two states. The core enzyme (β, β′ and two α subunits) is catalytically active but cannot bind DNA specifically[2]. Conversely, the holoenzyme contains a dissociable σ factor that delivers RNA polymerase to promoters[3]. Small accessory subunits, which vary between species, can be components of either core or holoenzyme but are not essential[4]. Housekeeping σ factors, found in all bacterial species, are essential and direct most transcription. Named $\sigma^{70}$ in *Escherichia coli*, and $\sigma^A$ in many other bacteria, housekeeping σ factors share four domains: $\sigma_1$, $\sigma_2$, $\sigma_3$ and $\sigma_4$[5]. The $\sigma_2$, $\sigma_3$ and $\sigma_4$ domains mediate promoter binding and DNA unwinding during transcription initiation[3,5]. Particularly important is the interaction between $\sigma_2$ and the promoter −10 element (consensus 5′-TATAAT-3′). This drives DNA opening that is the key prerequisite for transcription to begin[6]. Hence, amino acids in $\sigma_2$ that mediate DNA melting are conserved in diverse housekeeping σ factors[5]. Nevertheless, the kinetics of open complex formation vary between species and the reasons for this are not fully understood[7,8]. Once RNA polymerase holoenzyme has escaped the promoter, and entered the elongation phase of transcription, the σ factor dissociates[9]. Core enzyme then completes transcript synthesis before releasing the DNA template and nascent RNA. Reassociation with σ again directs RNA polymerase to a promoter and gene transcription begins afresh.

In addition to canonical genes and their promoters, most bacterial genomes contain large sections of horizontally acquired DNA[10]. These include remnants of prophages, pathogenicity islands, and conjugative elements[11,12]. Many such regions are conspicuous because their AT-content is higher than the surrounding genome (i.e. they are AT-rich)[10]. Thus, whilst encoded proteins can be beneficial, such loci can negatively impact fitness[13]. Working with *E. coli*, we recently showed that such defects can result from uncontrolled transcription[14,15]. In particular, sequences resembling promoter −10 elements occur frequently in AT-rich DNA, often within genes, and are used by $\sigma^{70}$ bound RNA polymerase to initiate transcription[14,16]. In turn, this sequesters RNA polymerase and canonical gene expression is reduced globally[15]. To avoid this, *E. coli* masks AT-rich DNA using Histone-like Nucleoid Structuring (H-NS) protein[17]. Present at 20,000 molecules per cell, H-NS forms filaments with stretches of AT-rich DNA to prevent promoter recognition and transcription elongation[10,15,18,19]. This process is known as xenogeneic silencing[20]. Genes encoding H-NS homologues are widely distributed in γ-proteobacteria[21]. Amongst the best characterised are the MvaT and MvaU proteins of *Pseudomonas aeruginosa*[22–24]. These factors also act to block spurious transcription initiation within horizontally acquired DNA[25]. Whilst the precise structural details differ, these silencers all recognise AT-rich sequences using arginine side chains that penetrate the DNA minor groove[23,26].

Beyond the γ-proteobacteria, most prokaryotes do not encode H-NS homologues. However, unrelated functional analogues have been identified. For example, in actinobacteria, Lsr2 uses a mode of DNA recognition resembling H-NS to silence foreign genes and non-coding antisense transcription[26,27]. In Gram-positive bacteria, the Rok protein of *B. subtilis* is best understood[28,29]. First identified as a repressor of natural competence[30], Rok was subsequently observed to bind parts of the *B. subtilis* genome that are AT-rich and foreign in origin[28]. Like H-NS, Rok can oligomerise and bind AT-rich DNA using its N- and C-terminal domains respectively[29]. However, nothing is known about mechanisms of silencing by Rok or the type of transcript initiation events targeted. It is noteworthy that Rok, present at ~1500 molecules per cell, is much less abundant than H-NS in *E. coli*[28].

Conversely, the AT-content of the *B. subtilis* genome, and horizontally acquired sequences within, is higher[11,12]. Hence, the need to prevent spurious transcription initiation, within foreign *B. subtilis* genes, would seem acute.

In this work, we have compared the xenogeneic silencing strategies of H-NS and Rok in *E. coli* and *B. subtilis* respectively. We show that, in sharp contrast to H-NS, Rok does not repress spurious intragenic transcription initiation. Instead, Rok specifically targets mRNA synthesis from canonical promoters. The different approaches reflect housekeeping RNA polymerase behaviour in each organism; the *E. coli* enzyme is more promiscuous, largely by virtue of the $\sigma^{70}$ factor. Thus, the expression of $\sigma^{70}$ in *B. subtilis* results in spurious intragenic transcription that Rok is unable to prevent. Conversely, *B. subtilis* $\sigma^A$ imposes greater specificity on *E. coli* RNA polymerase. Genome-wide, the two σ factors direct RNA polymerase to slightly different promoter sequences and *E. coli* $\sigma^{70}$ is less fastidious. We conclude that RNA polymerase specificity is key to the xenogeneic silencing strategy in *B. subtilis*. Conversely, transcriptional promiscuity is a barrier that must be overcome to silence horizontally acquired genes in *E. coli*.

## Results

**High-resolution mapping of the H-NS regulated transcriptome in *Escherichia coli*.** Previously, we used poly(5′-phosphatase)-sequencing to map *E. coli* transcription start sites (TSSs) genome-wide in the presence and absence of H-NS[14]. Cappable-seq offers numerous advantages over our prior approach[31]. Most notably, cappable-seq is more sensitive and specifically identifies the triphosphorylated 5′ ends of primary transcripts[31]. Hence, we began by remapping H-NS regulated TSSs in *E. coli* using cappable-seq. Our detection of transcription initiation events increased threefold (Supplementary Data 1). Parallel RNA-seq assays quantified transcript abundance (Supplementary Data 2) and H-NS ChIP-seq profiles[32] completed a high-resolution view of the regulon. A representative chromosomal region, which typifies the transcriptional response to H-NS, is shown in Fig. 1a. In wild-type cells, we observed low levels of transcription (red track) and few TSSs (pink track) across regions bound by H-NS (green track). By contrast, in the absence of H-NS, transcript abundance increased (blue track) and many TSSs were detected (mauve track). Consistent with our prior work, most of these H-NS repressed TSSs were intragenic[14,15]. The results are summarised as volcano plots in Fig. 1b. The top panel shows transcript abundance determined by RNA-seq and each data point represents an individual gene. Transcription across H-NS bound genes (green) is derepressed when H-NS is absent. In the lower panel, each data point corresponds to a TSS identified using cappable-seq; a similar trend is evident. Figure 1c illustrates the number of all *E. coli* TSSs locating to intragenic or intergenic DNA. As expected, for H-NS bound regions, more intragenic TSSs are found when H-NS is absent.

**High-resolution mapping of the Rok regulated transcriptome in *Bacillus subtilis*.** Like H-NS analogues in other organisms, the *B. subtilis* Rok protein recognises horizontally acquired AT-rich DNA[33]. In particular, Rok preferentially binds DNA sequences with TpA steps, rather than continuous A or T tracts[29,33]. Such sequences are ideal templates for transcription initiation events[14,15,34]. We next sought to understand how Rok might intervene and applied the approaches described above. The cappable-seq and RNA-seq results are described in Tables S1 and S2 respectively and were compared to existing ChIP-seq data for Rok binding[35]. A region of the *B. subtilis* genome with a typical transcriptional response to Rok is shown in Fig. 1d. As expected, in the presence of Rok, we observed low levels of transcription (red track) and infrequent TSSs (pink track) across the Rok

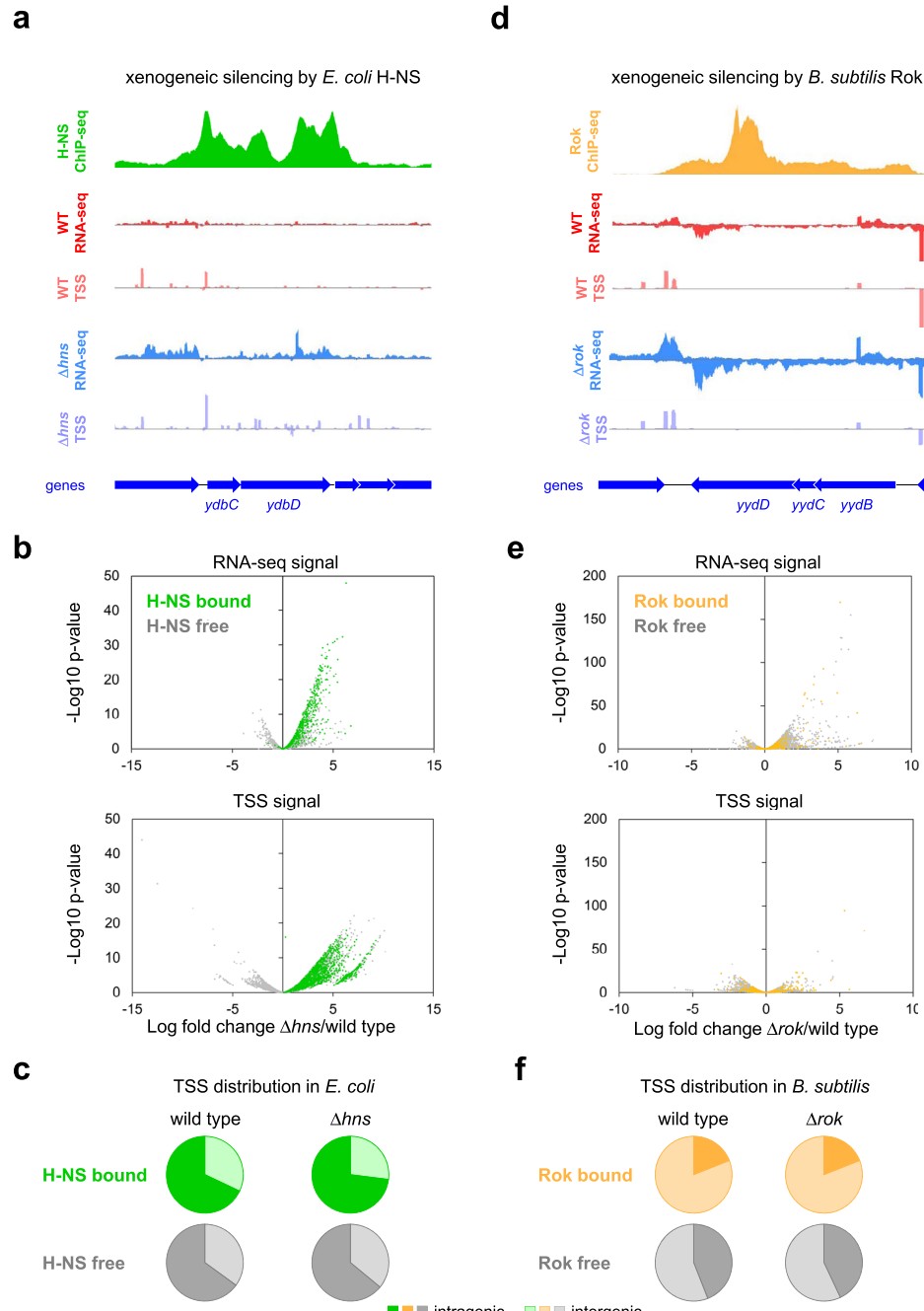

bound DNA (orange track). In the absence of Rok, levels of transcription increased (blue track) but additional TSSs were not detected (mauve track). The complete data are visualised as volcano plots in Fig. 1e. Levels of transcription across most Rok bound genes increased in the absence of Rok (top panel). Conversely, Rok had little influence on cappable-seq signals at TSSs (bottom panel). Genome-wide, TSSs were less likely to be intragenic in Rok bound regions and deletion of Rok had no impact on this distribution (Fig. 1f). Hence, despite the high sensitivity of cappable-seq, we did not detect widespread intragenic transcription initiation in the absence of Rok in *B. subtilis*. This differs markedly to the situation in *E. coli* for H-NS (Fig. 1a–c)[14,15].

**H-NS and Rok have different patterns of DNA binding in vivo.**
Whilst examining the ChIP-seq data we noticed differences in the distribution of H-NS and Rok (Figs. S1a and S1b). Most notably,

H-NS binding appeared to invade coding sequences more frequently. We quantified this apparent difference in two ways. First, we generated aggregate plots of H-NS and Rok binding, relative to gene start codons, for the complete ChIP-seq datasets. The results are shown in Supplementary Fig. 1c. For Rok, DNA binding is usually centred near gene start codons and tends not to encroach substantially on coding sequences (left panel). Conversely, for H-NS, binding peaks are much broader and encompass large tracts of coding DNA (right panel). Second, we examined the relationship between the binding signal intensity in ChIP-seq experiments and the occurrence of non-coding DNA. The results are shown in Supplementary Fig. 1d. For Rok, increasing the binding signal positively correlates with the occurrence of non-coding DNA. Hence, nearly all of the highest signals for Rok binding located to non-coding sequences (orange line). The pattern for H-NS is more complicated. The likelihood of DNA being non-coding is elevated until 40% of the maximum H-NS

**Fig. 1 Xenogeneic silencing impacts transcription differently in *E. coli* and *B. subtilis*. a** Repression of transcription by H-NS in *E. coli*. The genomic region encompassing *ydbCD* is shown. Data for H-NS occupancy are shown by the green graph[32]. The total RNA abundance determined by RNA-seq in wild-type and Δ*hns* cells is shown by the red and blue graphs respectively. Transcription start sites (TSSs) were identified by cappable-seq for wild-type (pink graph) and Δ*hns* (mauve graph) cells. In the cappable-seq data only RNA 5′ ends are sequenced and so the upstream edge of each peak indicates a TSS. Sequence reads mapping to the top and bottom DNA strands are shown above and below the central horizontal line in each plot. The y-axis scales are identical for data obtained using wild-type and Δ*hns* cells for each type of experiment. Genes are shown by blue arrows. **b** Volcano plots illustrating differences in the distribution of signals obtained by RNA-seq (top panel) or cappable-seq (bottom panel) in the presence and absence of H-NS in *E. coli*. For the RNA-seq analysis, each data point represents the average signal across an individual gene. In the cappable-seq data plot, each data point represents a separate TSS. For both plots, data points are coloured to indicate DNA regions bound by (green) or free from (grey) H-NS. **c** The pie charts illustrate the distribution of TSSs obtained by cappable-seq from wild-type (left hand side) and Δ*hns* (right hand side) *E. coli* cells. The TSSs are further separated into those in H-NS bound (green) and H-NS free (grey) regions. For all pie charts, dark shading indicates TSSs in coding DNA whilst pale shading identifies TSSs in non-coding regions. **d** Repression of transcription by Rok in *B. subtilis*. The genomic region encompassing *yydBCD* is shown. Data for Rok occupancy are shown by the orange graph[35]. Colour coding is otherwise as shown in (**a**) except that here the comparison is between wild-type and Δ*rok B. subtilis* cells. **e** Volcano plots illustrating differences signals obtained by RNA-seq (top panel) or cappable-seq (bottom panel) in the presence and absence of Rok in *B. subtilis*. Data points are as described for (**b**) and coloured to indicate DNA regions bound by (orange) or free from (grey) Rok. **f** Pie charts illustrate the distribution of TSSs obtained by cappable-seq from wild-type (left hand side) and Δ*hns* (right hand side) *B. subtilis* cells. The TSSs are further separated into those in Rok bound (orange) and Rok free (grey) regions. For all pie charts, dark shading indicates TSSs in coding DNA whilst pale shading identifies TSSs in non-coding regions.

binding signal is reached. Hence, regions of maximal H-NS binding are often within coding DNA.

***Bacillus subtilis* and *Escherichia coli* RNA polymerase have different propensities for spurious intragenic transcription in vitro**. We reasoned that a *B. subtilis* factor other than Rok might prevent transcription initiation, within AT-rich genes, in vivo. If correct, the *B. subtilis* RNA polymerase holoenzyme should generate spurious intragenic transcripts from naked DNA templates in vitro. The genomic region that we selected as an in vitro template is shown in Fig. 2a. The DNA segment includes *comK*, a well-characterised Rok target gene[30], and the upstream regulatory DNA. As expected, the *comK* gene is subject to direct repression by Rok in vivo (compare red and blue tracks in Fig. 2a). However, consistent with the observations described above, we did not detect intragenic transcription initiation in the absence of Rok (compare pink and mauve tracks in Fig. 2a). To measure transcription from this DNA locus in vitro we cloned *comK*, and the associated regulatory DNA, upstream of the λ*oop* terminator in plasmid pSR (see schematic in Fig. 2b). In this context, RNA polymerase is expected to generate a 696 nucleotide (nt) long mRNA from the *comK* promoter. Spurious RNAs of intragenic origin, terminated by the λ*oop* signal, should be smaller. The result of the in vitro transcription assay is shown below the Fig. 2b schematic. As expected, a 696 nt transcript was detected (lane 1) and subject to repression by Rok (lanes 2 and 3). However, we did not detect smaller transcripts initiating from sites within the *comK* gene. Transcripts >1000 nt in size result from transcription units elsewhere in the pSR plasmid. The RNAI transcript is encoded at the plasmid replication origin. The higher RNAI signal upon Rok addition likely results from more RNA polymerase being available when transcription elsewhere, including RNAs encoded by the plasmid backbone, is reduced. Our observation that *B. subtilis* RNA polymerase holoenzyme did not generate transcripts from sites within the *comK* coding sequence was surprising. Hence, in a control experiment, we re-examined the propensity of *E. coli* RNA polymerase holoenzyme to drive spurious transcription. Importantly, we selected an H-NS targeted gene from *E. coli* that had an AT-content of 56%, identical to that of *comK*. Consistent with our expectations, H-NS suppressed transcription from sites within the *agaB* coding sequence in vivo (Fig. 2c). A similar observation was made in vitro (Fig. 2d). Taken together, our data suggest that fundamentally different strategies are needed to ensure specific transcription of horizontally acquired DNA in *B. subtilis* and *E. coli*.

In particular, the *E. coli* RNA polymerase holoenzyme is intrinsically promiscuous. Hence, H-NS must block both intragenic transcription and mRNA production. Conversely, *B. subtilis* relies on enhanced RNA polymerase specificity to avoid transcription within genes. We next wanted to understand the basis for these differences.

**The *Bacillus subtilis* RNA polymerase uses a narrower range of promoter configurations and is more sensitive to discriminator sequence in vivo**. Our attention returned to our cappable-seq data. For each TSS detected we identified the associated promoter −10 element. The histograms in Fig. 3a show the distribution of distances separating TSSs from −10 elements. Consistent with many prior reports, the preferred distance was 7 bp for both *B. subtilis* and *E. coli*[36–38]. However, the overall distribution of distances was subtly different. In particular, for *E. coli*, more −10 elements were separated from TSSs by 8 or 6 bp. Next, we aligned all DNA sequences upstream of TSSs for each organism. The alignments were anchored by TSS position and are illustrated as DNA sequence logos in Fig. 3b. There are three notable differences between the logos derived for *B. subtilis* and *E. coli*. First, because −10 elements are less consistently positioned, the 5′-TATAAT-3′ consensus sequence is misrepresented for *E. coli*. Second, there is a clear preference for an AT-rich discriminator sequence in *B. subtilis*. Finally, the overall information content of the *E. coli* sequence logo is lower.

**The *Bacillus subtilis* RNA polymerase is sensitive to discriminator sequence and length in vitro**. Whilst informative, the DNA sequence logos in Fig. 3b represent aggregate properties of many promoters. We wanted to understand the impact of identified sequence features at a specific promoter. The *veg* promoter has been widely used as a model to study *B. subtilis* transcription[39]. Hence, we made derivatives of the *veg* promoter with different discriminator sequence (AT- or GC-rich) and length (Fig. 3c). A 155 nt transcript is generated from these promoters in our in vitro transcription system. Figure 3d shows a representative gel image from such experiments and a quantification from three independent assays. Transcription using the *B. subtilis* σ[A] holoenzyme is 2.5-fold lower if the AT-rich discriminator is replaced with a GC-rich sequence. The activity was reduced twofold further if the length of the GC-rich discriminator was increased by 1 base pair. Conversely, *E. coli* σ[70] RNA polymerase had indistinguishable activity at the different promoter

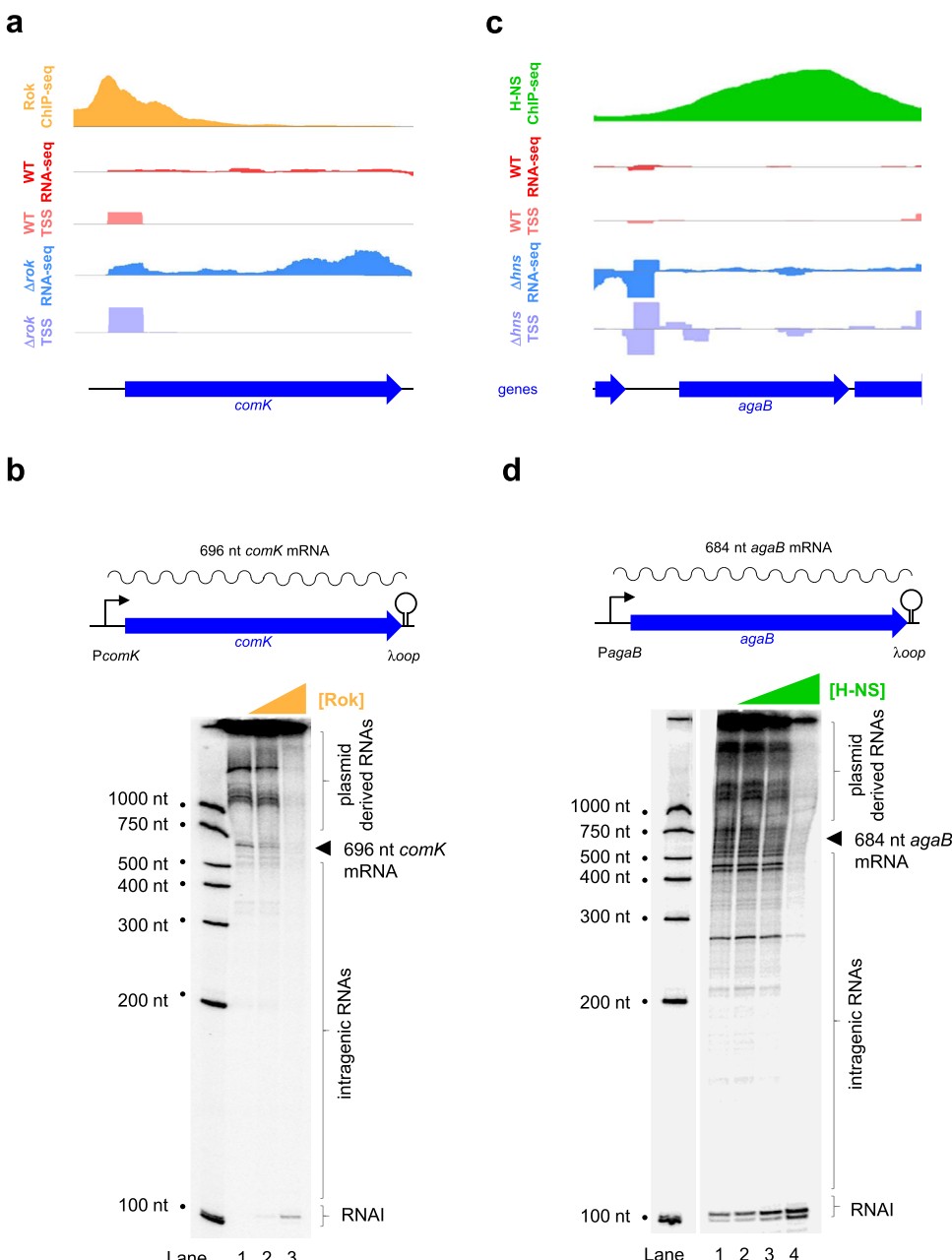

**Fig. 2 Housekeeping RNA polymerases of _E. coli_ and _B. subtilis_ differ in their promiscuity. a** Rok represses transcription of the _B. subtilis comK_ mRNA in vivo. Data for Rok occupancy (orange)[35], total RNA abundance (red and blue) and transcription start site (TSS) usage (pink and mauve) are shown. Sequence reads mapping to the top and bottom DNA strands are shown above and below the central horizontal line in each plot. The _y_-axis scales are identical for data obtained using wild-type and Δ_rok_ cells for each type of experiment. Genes are shown by blue arrows. **b** Rok represses transcription of the _B. subtilis comK_ mRNA in vivo. The schematic illustrates the _comK_ gene and regulatory region, cloned in plasmid pSR, and used as a template for in vitro transcription. The _comK_ TSS is shown as a bent black arrow, the _comK_ gene is shown as a block blue arrow, and the sequence encoding the λ_oop_ transcriptional terminator is indicated by a stem loop schematic. _B. subtilis_ σ^A RNA polymerase (0.5 μM) and Rok (0, 0.5, or 1.0 μM) were added as indicated. Note that the 696 nt _comK_ mRNA is easily discernible and there is no evidence for transcription initiation within _comK_. Species of RNA over ~1000 nt in length are derived from sites elsewhere on the plasmid template. The RNAI transcript is encoded by the plasmid replication origin. The experiment was done twice with similar results. **c** H-NS represses transcription initiation within the _E. coli agaB_ coding sequence in vivo. Data for H-NS occupancy are in green[32] and otherwise as indicated in (**a**) except that wild-type and Δ_hns E. coli_ cells are compared. **d** H-NS represses transcription initiation within the _E. coli agaB_ coding sequence in vitro. The schematic illustrates a section of DNA cloned in plasmid pSR and used as a template for in vitro transcription. The expected size of the _agaB_ mRNA is 684 nucleotides (nt). The gel image shows transcripts generated by _E. coli_ σ^70 RNA polymerase (0.5 μM) using this DNA template. The expected position of _agaB_ mRNA is indicated by an arrow head but is obscured by many similarly sized and smaller transcripts derived from _agaB_ coding sequence. H-NS was added at concentrations of 0, 0.5, 1.0 or 2.0 μM. The experiment was done twice with similar results.

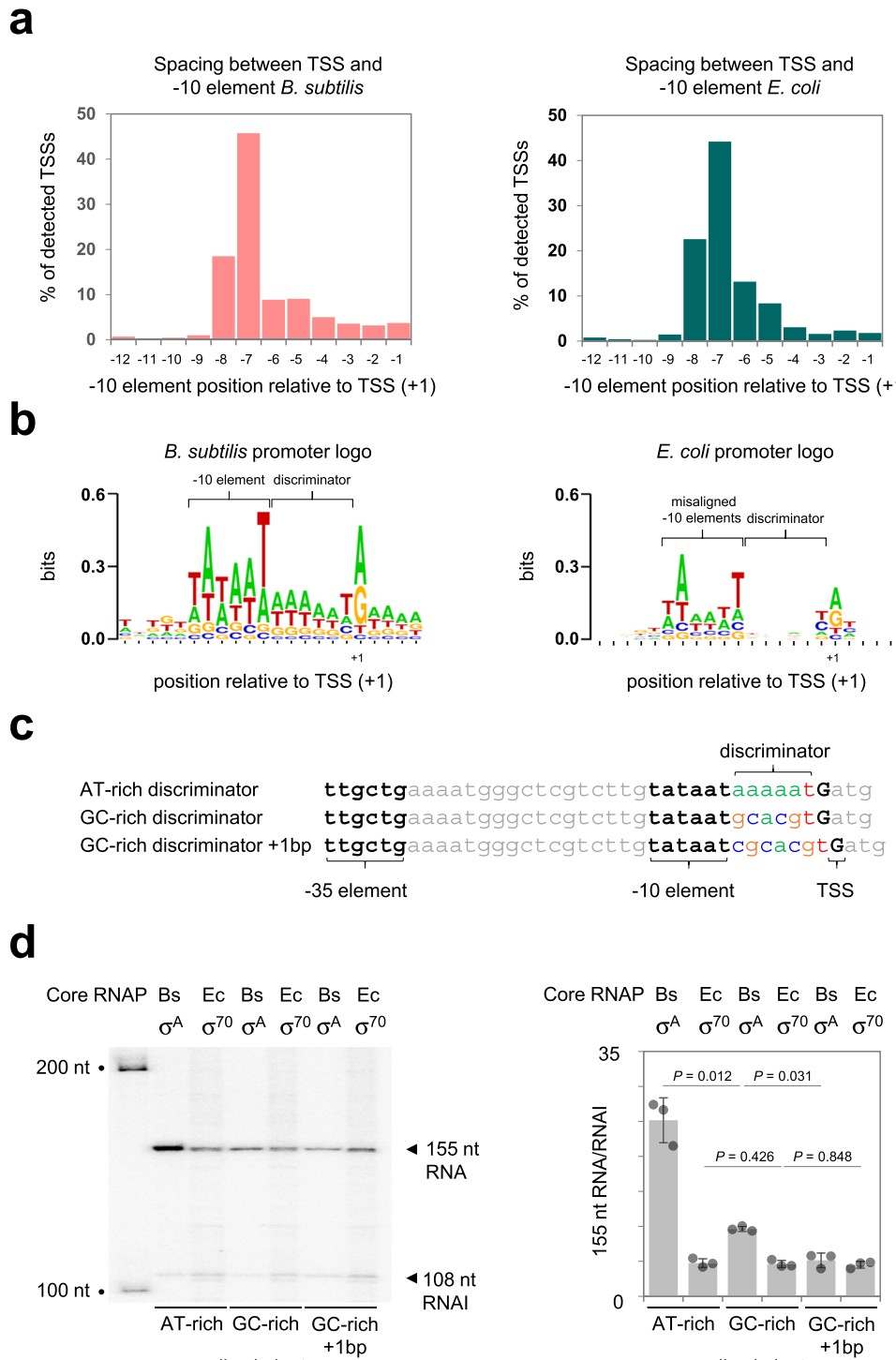

**Fig. 3 *E. coli* housekeeping RNA polymerase better tolerates promoter sequence variation. a** Positioning of promoter −10 elements and transcription start sites in *E. coli* and *B. subtilis*. The bar charts show the percentage of promoter −10 elements located at indicated distances upstream of transcription start sites (TSSs, +1) identified by cappable-seq for *E. coli* and *B. subtilis*. **b** The panel shows DNA sequence logos generated by aligning nucleic acid regions upstream of *B. subtilis* (left) or *E. coli* (right) transcription start sites. The more variable spacing between transcription start sites and promoter −10 elements in *E. coli* generates a motif that misrepresents the consensus −10 element sequence (5′-TATAAT-3′). There is no overall sequence preference for the promoter discriminator region in *E. coli* whilst an AT-rich sequence is common in *B. subtilis*. **c** DNA sequences of the *B. subtilis veg* promoter and derivatives with either an AT-rich or GC-rich discriminator sequence. **d** Results of in vitro transcription assays using DNA templates containing one of the promoter sequences shown in (**c**). Experiments were done with either the *B. subtilis* (Bs) σ^A, or *E. coli* (Ec) σ^70, RNA polymerase holoenzyme (0.5 μM). The 155 nucleotide (nt) transcript is generated from the cloned promoter and the 108 nt RNAI transcript is derived from the plasmid replication origin. The gel image shows a representative result and the bar chart shows a quantification of three independent experiments. Error bars show S.D. and the centre of the error bars is the mean. *P* was calculated using a two-tailed student's *T*-test.

sequences. Supplementary Fig. 2 shows results of $KMnO_4$ footprints that detect DNA melting. Open complexes formed by the *B. subtilis* enzyme are notoriously unstable and could not be detected using our assay[7,40,41]. Unwinding dependent on *E. coli* $\sigma^{70}$ was detected and less stable with the GC-rich discriminator.

**Escherichia coli $\sigma^{70}$ directs intragenic transcription of Rok bound genes by *Bacillus subtilis* core RNA polymerase in vivo.** Our observations are consistent with the housekeeping *B. subtilis* RNA polymerase using a narrower range of promoter configurations than the equivalent *E. coli* enzyme. This may explain the different behaviour of each RNA polymerase at horizontally acquired genes. RNA polymerase-promoter interactions are primarily mediated by the exchangeable $\sigma$ subunit[3]. Indeed, prior studies have shown that foreign $\sigma$ factors, expressed in a given bacterium, can drive the host RNA polymerase to different promoters[42–44]. Hence, we reasoned that *E. coli* $\sigma^{70}$ might render *B. subtilis* core RNA polymerase more promiscuous. To test this prediction, the $\sigma^{70}$ encoding *rpoD* gene was cloned in plasmid pDR111. The resulting DNA construct was used to transform *B. subtilis* strain 168ca. Following transformation, pDR111 integrates with the *B. subtilis* chromosome at the *amyE* locus. Hence, transformants encode a single copy of *rpoD* under the control of an IPTG inducible promoter. We used cappable-seq to identify TSSs in transformed cells. We compared these TSSs to those detected in the absence of $\sigma^{70}$. In total, the two cappable-seq analyses identified 8524 TSSs (Fig. 4a, Supplementary Data 1). The vast majority of these are likely to be dependent on either $\sigma^{70}$ or $\sigma^{A}$ rather than an alternative $\sigma$ factor. Consistent with this, 82% of the TSSs were correctly positioned downstream of a housekeeping promoter −10 element. Of all TSSs, 2510 were identified only in the presence of $\sigma^{70}$. These likely represent otherwise silent promoters that can be used only when $\sigma^{70}$ is expressed. There were 4593 TSSs identified both in the presence and absence of $\sigma^{70}$. The associated promoters could be used by both $\sigma^{70}$ and $\sigma^{A}$. Alternatively, some of these promoters could use $\sigma^{A}$ specifically, even if $\sigma^{70}$ is present. Finally, we identified 1421 TSSs only in the absence of $\sigma^{70}$. Such TSSs are likely to be $\sigma^{A}$ specific but not used when $\sigma^{70}$ is expressed. For instance, $\sigma^{70}$ could direct the limited pool of core RNA polymerase to alternative locations. Recall that our analysis above-identified subtle differences between *E. coli* and *B. subtilis* promoter sequences (Fig. 3a, b). Consistent with this, DNA sequence logos generated from *E. coli* $\sigma^{70}$ dependent promoters found in *B. subtilis* had misaligned −10 elements and less AT-rich discriminators (Supplementary Fig. 3a). Conversely, canonical *B. subtilis* promoters did not exhibit these sequence features (Supplementary Figs. 3b, c). Most importantly, TSSs dependent on $\sigma^{70}$ were more abundant in AT-rich islands targeted by Rok (compare left and right charts in Fig. 4b). Some examples are shown in Fig. 4c. In parallel experiments, Rok was similarly unable to silence intragenic transcription in *E. coli* cells lacking H-NS (Supplementary Fig. 4)

**Escherichia coli $\sigma^{70}$ directs intragenic transcription of Rok bound genes by *Bacillus subtilis* core RNA polymerase in vitro.** We expected that swapping $\sigma^{70}$ and $\sigma^{A}$, in the context of the *E. coli* and *B. subtilis* RNA polymerase holoenzymes, would alter transcriptional promiscuity in vitro. We tested this using in vitro transcription assays. The DNA templates were six different *B. subtilis* genes that we had identified as being targeted by Rok. None of these genes were subject to promiscuous transcription by *B. subtilis* RNA polymerase in cells lacking Rok. Lanes 1–6 show transcripts generated by *B. subtilis* core RNA polymerase in conjunction with $\sigma^{A}$ (Fig. 4d). Note that *yydD* is located within an operon and so there is no regulatory upstream DNA. For all of

the other DNA templates the expected mRNA species were detected (marked by coloured triangles). Conversely, many intragenic transcripts were generated when the *B. subtilis* core enzyme was provided with $\sigma^{70}$ (Lanes 7–12). Equivalent experiments using the *E. coli* core RNA polymerase are shown in Lanes 13–24. Again, promiscuity was dictated by the provided $\sigma$ factor.

**Side chains R156 and R486 contribute to the promiscuity of *E. coli* $\sigma^{70}$ and are absent in *B. subtilis* $\sigma^{A}$.** Both $\sigma^{70}$ and $\sigma^{A}$ comprise four highly conserved domains (named $\sigma_1$, $\sigma_2$, $\sigma_3$ and $\sigma_4$) (Fig. 5a). The *E. coli* $\sigma^{70}$ factor contains a further determinant, known as the non-conserved region (NCR), not present in $\sigma^{A}$. The aligned amino acid sequences of the $\sigma$ factors are shown in Supplementary Fig. 5. Strikingly, amino acids responsible for interaction with core RNA polymerase and core promoter elements, at each step of transcription initiation, are near identical (Supplementary Fig. 5). Hence, important differences must be subtle. In an initial effort to compare the two proteins we made a series of hybrid $\sigma$ subunits with different domain combinations. However, in all cases, these hybrid $\sigma$ factors were poorly active. Hence, in a more nuanced approach, we examined changes between $\sigma^{70}$ and $\sigma^{A}$ in the context of the *E. coli* RNA polymerase structure[3,45]. In particular, we searched for differences in surface-exposed amino acid side chains that might alter interactions with core RNA polymerase or the promoter DNA. This identified two residues of interest. Side chain R157 of $\sigma^{70}$ is located in the NCR and so absent from $\sigma^{A}$. In the context of the closed holoenzyme-promoter complex, R157 is 14.5 Å from the DNA backbone. Conformational changes during DNA opening place R157 just 4.1 Å from the DNA. Narayanan et al. previously showed that R157 contacts the DNA backbone in this context to impact DNA opening[45]. Side chain R486 of $\sigma^{70}$ is located in region 3.2 of domain $\sigma_3$. In the closed complex R486 is 10.6 Å from the DNA and moves to a more distal 22.6 Å in the open complex. In *B. subtilis*, $\sigma^{A}$ residue D222, corresponding to $\sigma^{70}$ R486, is negatively charged. We altered $\sigma^{70}$, with mutations R157A and R486D, to more closely resemble $\sigma^{A}$, and refer to this derivative as $\sigma^{70\ \mathrm{Mut}}$. First, we compared the properties of $\sigma^{70}$, $\sigma^{A}$, and $\sigma^{70\ \mathrm{Mut}}$ at the *B. subtilis veg* promoter using either $KMnO_4$ footprinting (to detect DNA opening) or in vitro transcription assays. Each $\sigma$ factor was tested with the core RNA polymerases of both *B. subtilis* and *E. coli*. As expected, holoenzymes containing the *B. subtilis* core RNA polymerase formed less stable open complexes than those containing *E. coli* core enzyme (compare lanes 1–3 with 4–6 in Fig. 5c). We observed subtle differences in patterns of DNA opening for $\sigma^{70}$ and $\sigma^{A}$. For instance, when associated with *B. subtilis* core RNA polymerase, two sites of $KMnO_4$ reactivity were observed for each $\sigma$ factor. These were at promoter positions -1 and -3 (lanes 1–3). Compared to $\sigma^{70}$, $\sigma^{A}$ resulted in greater $KMnO_4$ reactivity at promoter position -3 (compare lanes 1 and 2). In this context, $\sigma^{70\ \mathrm{Mut}}$ behaviour resembled $\sigma^{A}$ rather than $\sigma^{70}$ (lane 3). Whilst more extensive DNA opening was observed in the context of the *E. coli* core RNA polymerase the different $\sigma$ factors again resulted in different $KMnO_4$ reactivity at the -3 position (lanes 4–6). However, in this instance, $\sigma^{70}$ generated a greater -3 signal (lane 5). Again, $\sigma^{70\ \mathrm{Mut}}$ behaved like $\sigma^{A}$ rather than $\sigma^{70}$. We also observed differences in the amount of transcript generated by RNA polymerase with each $\sigma$ factor (Fig. 5d). There were significant differences in behaviour $\sigma^{A}$ and $\sigma^{70}$ but not between $\sigma^{A}$ and $\sigma^{70\ \mathrm{Mut}}$.

**Mutation of $\sigma^{70}$ side chains R156 and R486 improves specificity at high AT-content DNA templates.** To determine if $\sigma^{70\ \mathrm{Mut}}$ directed less promiscuous transcription we compared $\sigma$ factor

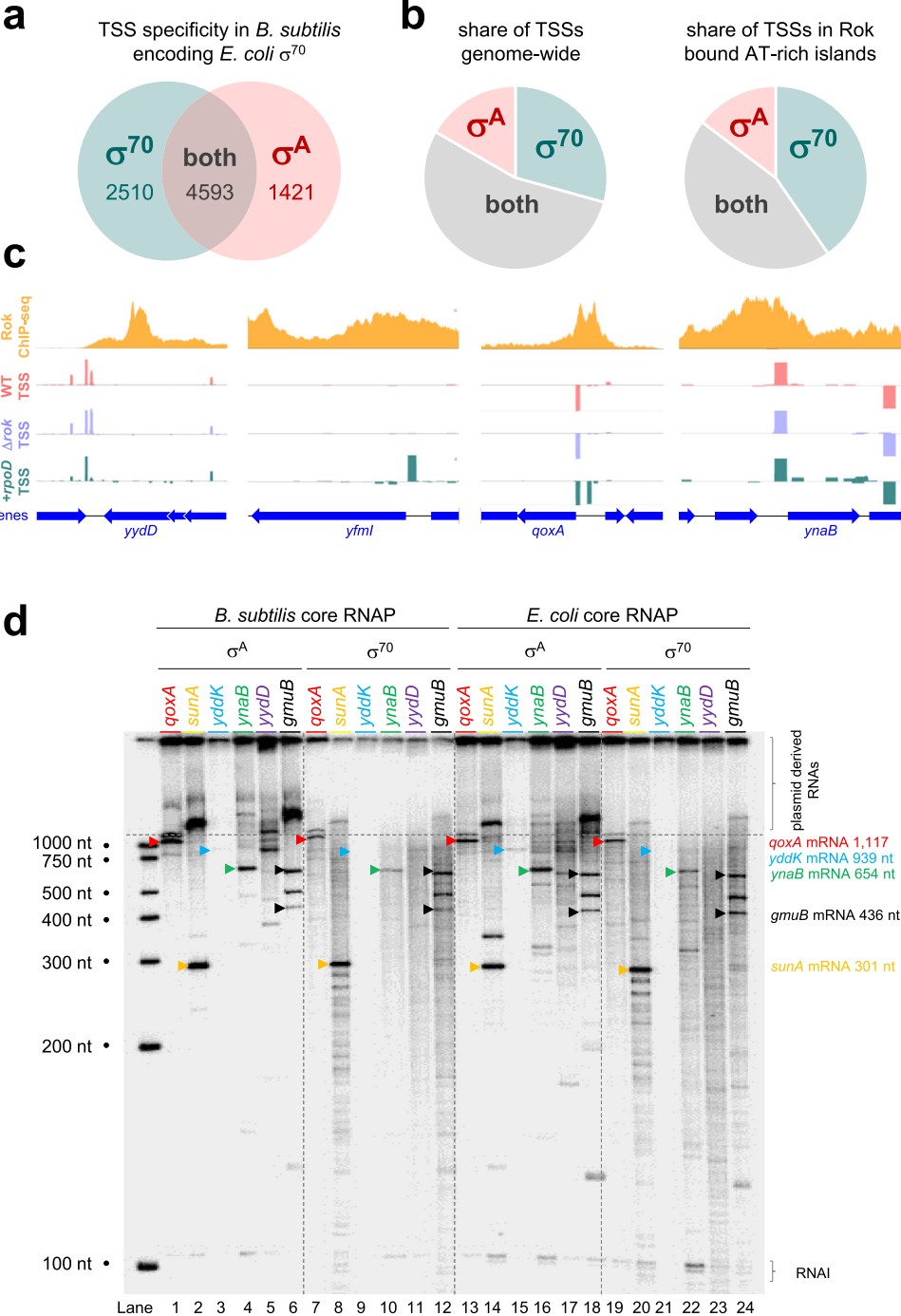

**Fig. 4 Expression of the *E. coli* RNA polymerase σ70 subunit is sufficient for promiscuous transcription in *B. subtilis*. a** The Venn diagram shows the distribution of TSSs identified in *B. subtilis* by cappable-seq in different genetic backgrounds. The teal area represents TSSs only detected upon expression of *E. coli rpoD* whilst the pink section represents TSSs only identified in the absence of *rpoD* expression. The overlap identifies those TSSs detected both with and without *rpoD* expression. **b** The pie charts show the distribution of *B. subtilis* TSSs identified in different genetic backgrounds and in different parts of the genome. The number of TSSs dependent on *E. coli* σ70 expression is higher in horizontally acquired AT-rich sections of DNA targeted by Rok. Conversely, the number of σA dependent TSSs is lower in these regions. **c** Examples of *E. coli* σ70 dependent transcription initiation within horizontally acquired *B. subtilis* genes. Data from ChIP-seq experiments for Rok occupancy[35] are shown by the orange graph. Transcription start sites (TSSs) were identified by cappable-seq for wild-type (pink graph), Δ*rok* (mauve graph) and *B. subtilis* cells carrying the σ70 encoding *rpoD* gene (teal graph). Sequence reads mapping to the top and bottom DNA strands are shown above and below the central horizontal line in each plot. Genes are indicated with block blue arrows. **d** Results of in vitro transcription assays using AT-rich horizontally acquired genes targeted by Rok cloned in plasmid pSR as the DNA template. Transcription reactions were done using core RNA polymerase, from either *B. subtilis* or *E. coli*, in conjunction with either σA or σ70 (0.5 μM final holoenzyme concentration). For each DNA template, bands corresponding to full-length mRNAs are indicated by coloured arrowheads. The experiment was done twice with similar results.

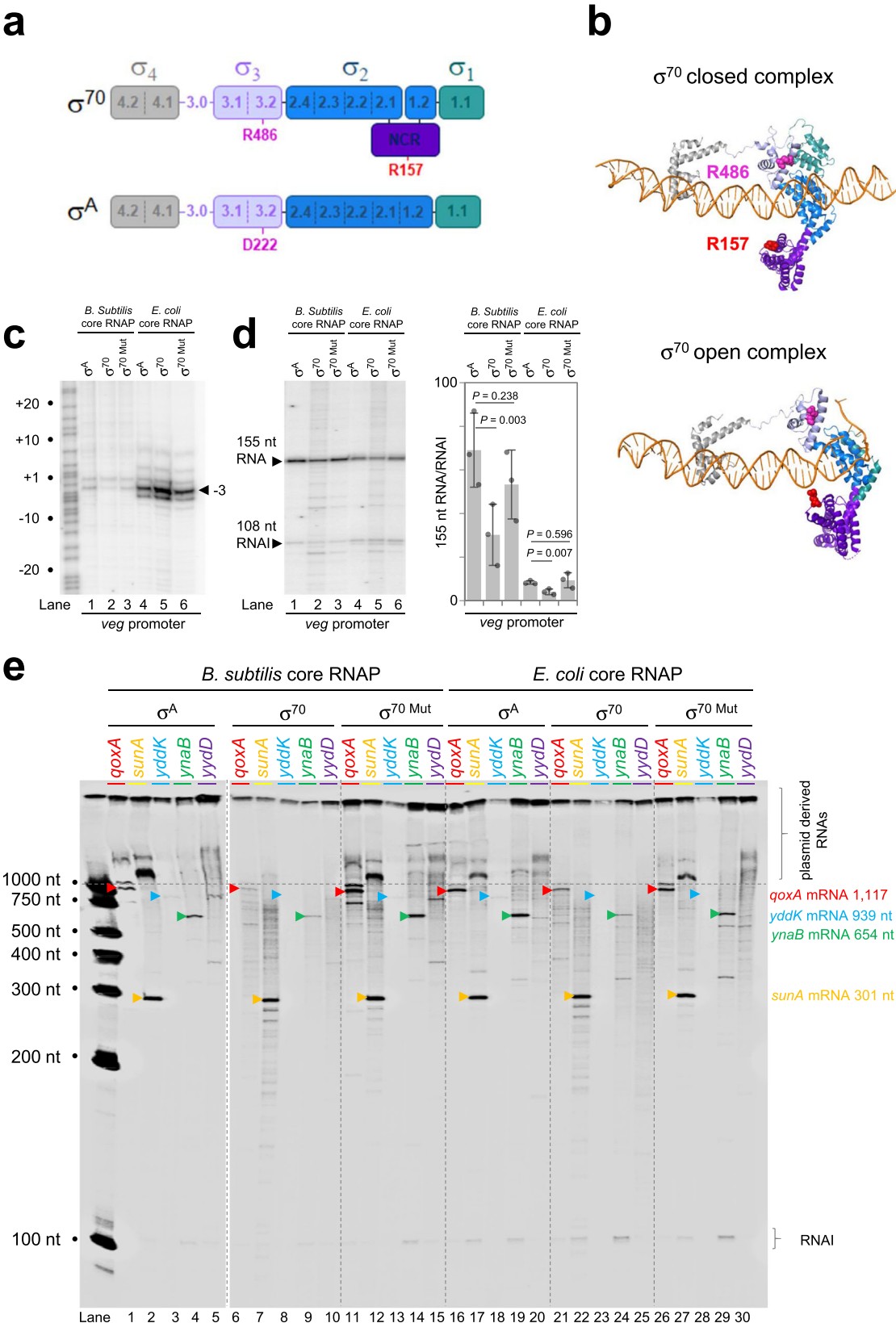

behaviour with DNA templates derived from Rok targeted genes (Fig. 5e). As described above, σ^A specifically stimulated the initiation of transcription at intergenic *B. subtilis* promoters to produce mRNAs (marked with coloured triangles, lanes 1–5 and 16–20). Conversely, σ^70 resulted in promiscuous initiation (lanes 6–10 and 21–25). The promiscuity of σ^70 Mut was reduced; we

detected greater production of expected mRNA species and reduced transcription of spurious RNAs (lanes 11–15 and 26–30). We conclude that σ^70 residues R156 and R486, which influence patterns of DNA opening, are important for the more promiscuous behaviour of core RNA polymerase associated with *E. coli* σ^70.

**Fig. 5 Mutation of *E. coli* σ⁷⁰ to resemble *B. subtilis* σᴬ reduces transcriptional promiscuity. a** Schematic representations of *E. coli* σ⁷⁰ and *B. subtilis* σᴬ. Individual domains are labelled σ¹ through σ⁴ and the non-conserved region (NCR) specific to *E. coli* σ⁷⁰ is also shown. Sub-regions of each σ factor are labelled 1.1 through 4.2 and are separated by dashed lines where required. Side chains R157 and R486 important for the promiscuous behaviour of σ⁷⁰ are shown. Side chain D222 of σᴬ is in the position equivalent to that of R486 in σ⁷⁰. **b** Location of R157 and R486 in *E. coli* σ⁷⁰ RNA polymerase bound to promoter DNA. The top and bottom images are derived from PDB accession numbers 6PSQ[3] and 6CA0[45] respectively. Whilst present in the structures, RNA polymerase core enzyme has been hidden from view for clarity. Colour coding of σ⁷⁰ matches (**a**) and DNA is shown in orange. **c** The *E. coli* σ⁷⁰ ᴹᵘᵗ derivative has DNA opening properties similar to *B. subtilis* σᴬ. The gel image shows KMnO₄ reactivity patterns at the *B. subtilis veg* promoter due to DNA opening by *B. subtilis* (lanes 1–3) or *E. coli* (lanes 4–6) core RNA polymerase in complex with either σᴬ, σ⁷⁰ or σ⁷⁰ ᴹᵘᵗ as indicated (0.5 μM final holoenzyme concentration). The gel is calibrated with a Maxam-Gilbert G + A sequencing reaction. The experiment was done twice with similar results. **d** Results of in vitro transcription assays using *B. subtilis* (lanes 1–3) or *E. coli* (lanes 4–6) core RNA polymerase in complex with either σᴬ, σ⁷⁰ or σ⁷⁰ ᴹᵘᵗ as indicated (0.5 μM final holoenzyme concentration). The 155 nucleotide (nt) transcript is generated from the *B. subtilis veg* promoter and the 108 nt RNAI transcript is derived from the plasmid replication origin. The gel image shows a representative result and the bar chart shows a quantification of 3 independent experiments. Error bars show S.D. and the centre of the error bars is the mean. *P* was calculated using a two-tailed student's *T*-test. **e** Results of in vitro transcription assays using AT-rich horizontally acquired genes targeted by Rok cloned in plasmid pSR as DNA templates. Transcription reactions were done using core RNA polymerase, from either *B. subtilis* or *E. coli*, in conjunction with either σᴬ, σ⁷⁰ or σ⁷⁰ ᴹᵘᵗ (0.5 μM final holoenzyme concentration). For each DNA template, bands corresponding to full-length mRNAs are indicated by coloured arrowheads. The experiment was done twice with similar results.

**The *B. subtilis* δ subunit also contributes to specific transcription of horizontally acquired genes.** The *B. subtilis* RNA polymerase contains two small ancillary subunits, named δ and ε, that are not found in *E. coli*[4]. Comparatively, the role of these subunits is poorly understood. However, previous work has implicated δ in control of transcriptional specificity and DNA opening[40,46–49]. Thus, in a final experiment, we remapped *B. subtilis* TSSs in the absence of δ. This identified 6786 TSSs and 1916 of these were unique to cells lacking the δ encoding *rpoE* gene. Interestingly, GTP was the most frequently used initiating nucleotide triphosphate (iNTP) at TSSs specific to the Δ*rpoE* strain (47% of TSSs). Conversely, ATP was the most common iNTP globally (52% of TSSs). This is intriguing since Rabatinová and co-workers implicated δ in the control of transcription by iNTP concentration[50]. There were no other sequence differences (Supplementary Fig. 6a). A greater proportion of TSSs in Rok targeted AT-rich islands were specific to Δ*rpoE* cells compared to the rest of the genome (Supplementary Fig. 6b). An example is shown in Supplementary Fig. 6c.

## Discussion

We propose fundamentally different xenogeneic silencing strategies in *E. coli* and *B. subtilis* (Fig. 6). In the former, RNA polymerase promiscuity creates a need for widespread transcriptional silencing by H-NS[14,15,18,34]. Hence, xenogeneic silencing inhibits both mRNA expression and spurious intragenic transcription initiation[14]. Conversely, Rok binds mostly to non-coding DNA and better RNA polymerase specificity reduces intragenic transcription (Figs. S1 and 5). We attribute differences in RNA polymerase promiscuity primarily to the housekeeping σ factor (Figs. 4 and 5). Consistent with this, Rok is unable to block intragenic transcription when *E. coli* σ⁷⁰ is expressed in *B. subtilis* (Supplementary Fig. 4). Similarly, compared to σ⁷⁰, σᴬ imposes greater specificity on core *E. coli* RNA polymerase (Figs. 4 and 5). Previous work, with a small number of model promoters, also noted the greater specificity of *B. subtilis* housekeeping RNA polymerase[38,51,52]. One such study concluded that core enzyme, not the associated σ factor, enforced greater specificity[42]. The latter conclusion was based on in vitro analysis of two phage promoters, λ PR' and T7A1. We do not exclude a role for core RNA polymerase in promoter selection related to transcriptional promiscuity. Indeed, core RNA polymerase has a clear influence on DNA opening in our assays (Fig. 5c). However, the ability of *E. coli* σ⁷⁰ to divert *B. subtilis* core enzyme to many different promoter sequences, both in vivo and in vitro, suggests σ must play a key role (Figs. 4 and 5). Consistent with prior

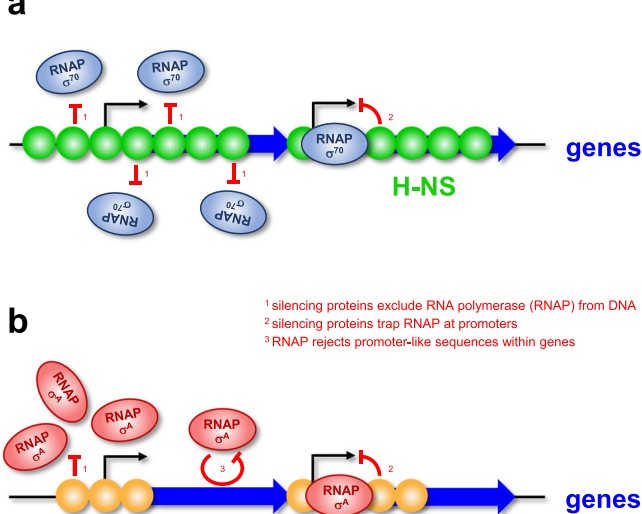

**Fig. 6 Different xenogeneic silencing strategies in *E. coli* and *B. subtilis*. a** In *E. coli*, H-NS (green) binds to extensive tracts of DNA. Consequently, the comparatively promiscuous *E. coli* housekeeping RNA polymerase (blue ovals) is prevented from synthesising mRNAs, and many spurious intragenic RNAs, from sections of AT-rich horizontally acquired DNA (blue block arrows). H-NS may repress transcription by blocking access of RNA polymerase to the DNA and by trapping RNA polymerase at promoters (bent arrows). **b** In *B. subtilis*, Rok (orange) binds to shorter tracts of DNA around promoters (bent arrows) at gene 5′ ends. This stops the synthesis of mRNAs from sections of AT-rich horizontally acquired DNA (blue block arrows). The less promiscuous housekeeping RNA polymerase of *B. subtilis* containing σᴬ (red ovals) is not prone to spurious intragenic transcription initiation. Trapping of RNA polymerase early during elongation appears commonplace.

reports, the *B. subtilis* δ subunit also contributes[7,53]. However, σ is dominant and impacts spurious intragenic promoters even when δ is present (Figs. 3 and 4). Our genome-scale TSS analyses identified sequence properties consistent with prior work[38,50,52]. For example, Henkin and Sonenshein noted that an AT-rich discriminator allowed the *E. coli lacUV5* promoter to be used by *B. subtilis* holoenzyme[51]. We speculate that greater specificity of *B. subtilis* RNA polymerase holoenzyme may have evolved to accommodate a higher AT-content genome in this organism.

Together, our cappable-seq and RNA-seq analyses suggest that Rok primarily silences mRNA transcription early during elongation; TSS signals for Rok silenced mRNAs are frequently similar both in the presence and absence of Rok. Conversely, there are substantial differences in signals for full-length mRNAs in RNA-seq experiments. Figures 1d and 2a both illustrate this behaviour that is also consistent with global patterns of transcription (Fig. 1e). In *E. coli*, H-NS can silence transcription both by excluding RNA polymerase from the DNA template and by trapping RNA polymerase at the promoter[54]. This property of H-NS reflects the protein's ability to switch between linear and bridging modes of DNA binding. Hence, we speculate that Rok may be less versatile in this regard. Consistent with this, biophysical measurements indicate that Rok is a DNA bridging protein[55].

The differences in promiscuity of *E. coli* and *B. subtilis* RNA polymerase are intriguing given the different role of the transcription termination factor Rho in these organisms. In *E. coli*, Rho is essential and reducing Rho activity results in greater spurious transcription across horizontally acquired genes. Numerous genetic interactions between *hns* and *rho* have been reported and are consistent with the factors working together to suppress unwanted transcription[56,57]. By sharp contrast, Rho is not essential in *B. subtilis*[58]. Indeed, loss of *rho* has only a minor impact on cell fitness[58]. In conclusion, natural selection has identified diverse strategies for managing horizontally acquired DNA in different bacteria. Most likely, the correct balance of silencing factors and transcriptional specificity is essential.

## Methods

**Strains, plasmids, and oligonucleotides**. Bacterial strains, plasmids, synthesised gene strands and oligonucleotides used in this study are listened in Supplementary Table 1. Bacterial cultures were grown at 37 °C in Lennox Broth (LB) medium. Cloning was done using HiFi Gibson assembly (New England Biolabs). Insert DNA was either synthesised as gene strands (Eurofins Genomics) or amplified from genomic DNA using oligonucleotides (Eurofins Genomics). The *B. subtilis rpoD* expression strain was constructed by cloning *E. coli rpoD*, with an optimal ribosome binding site, in plasmid pDR111. The resulting constructs were used to transform *B. subtilis* 168c and double crossover integration at the chromosomal *amyE* locus was confirmed by iodine testing patched cells grown on LB agar with 1 % (*w/v*) starch.

**Cappable-seq and RNA-seq**. All experiments were done twice using total RNA isolated from biological replicates. Strains were grown in LB with shaking at 37 °C until mid-log phase. Aliquots of 2 ml were then pelleted and flash frozen in liquid nitrogen. For expression of *rpoD* in *B. subtilis*, 3 mM IPTG (final) was added at exponential phase for 1 h before harvesting cells. For both *B. subtilis* and *E. coli* RNA-seq and cappable-seq experiments were done as described in Warman et al.[37]. Vertis Biotechnologie AG (Germany) did library preparation steps followed by sequencing with an Illumina NextSeq 500 system (75 bp read length). Raw data in FASTQ format are available from ArrayExpress (accession number E-MTAB-10777, https://www.ebi.ac.uk/arrayexpress/experiments/E-MTAB-10777/).

**Bioinformatics**. Individual sequence reads in FASTQ files were aligned to the reference genomes NC000964.3 (*B. subtilis*) or U00096.3 (*E. coli*) using Bowtie2 (Galaxy version 2.4.2)[59]. Coverage for each genome position was extracted from resulting Binary Alignment Map (BAM) files using the genomcov function of BedTools (Galaxy version 2.30.0)[60]. For each strand, coverage was used to call TSSs at positions where read depth increased more than threefold, compared to the previous base, in both replicates. Positions where the read depth was zero are excluded to avoid erroneous TSS selection. To generate DNA sequence logos, sequences upstream of the desired TSSs were aligned by TSS position and submitted to WebLogo (version 2.8.2)[61]. Standard settings were used. The R package GenomicRanges (version 1.44)[62] was used to identify which TSSs fell within different genomic contexts (e.g. H-NS or Rok bound regions) using the coordinates in Supplementary Data 3. FeatureCounts (version 2.6)[63] of the Rsubread (version 2.6)[64] package was used to determine gene read count, which were inputted into the exact function of edgeR (Galaxy version 3.34.0)[65] to determine differential gene expression. For differential TSS activity, TSSs for the two samples to be compared were pooled, with duplicate TSSs removed. The coverage at each TSS position on the genome was calculated for biological duplicates submitted to the exact function of edgeR[65]. Note that edgeR automatically adjusts for sequencing depth discrepancies when calculating fold-changes and *P* values.

To quantify differences in Rok and H-NS distribution we used existing ChIP-seq data[32,35]. Binding signals for each factor were averaged every 10 bp across the entire genome. The average ChIP-seq signal for all 10 bp bins was then calculated and this value was subtracted from each bin to remove background binding signals. The binned data were processed in two different ways. First, the distance between each 10 bp bin and the nearest gene start codon was determined. The sum of ChIP-seq binding signals was then calculated for all bins falling in a given range of distances upstream or downstream of a start codon (Supplementary Fig. 1c). Second, we determined the ChIP-seq signal for each bin as a percentage of the maximum ChIP-seq signal for H-NS or Rok. These values were compared to the percentage of bins locating to non-coding DNA in Supplementary Fig. 1d. Distances between promoter −10 elements and TSSs were determined as described previously[37].

**Proteins**. *B. subtilis* RNA polymerase was purified from a strain expressing a chromosomal β′ C-terminal domain His$_6$ tag fusion[66]. *B. subtilis* cells were grown at 37 °C with shaking to late exponential phase in LB medium supplemented with 1% (*w/v*) glucose. Cell pellets were resuspended in lysis buffer (50 mM Tris pH 7.9, 300 mM NaCl, 3 mM β Mercaptoethanol, 5% and cOmplete EDTA-free protease inhibit tablet) and lysed by sonication. Lysate was clarified by centrifugation at 45,000 × *g* for 20 min at 4 °C followed by filtration through a 0.45 μM PES filter. Lysates were applied to a His-Trap HP column (GE Healthcare). Unbound protein was removed by washing the column with lysis buffer followed by lysis buffer with 25 mM imidazole. Bound protein was eluted in a gradient to 200 mM imidazole in lysis buffer. RNA polymerase containing fraction eluted between 120 and 150 mM imidazole. These fractions were pooled, diluted 3-fold in HiTrap buffer (40 mM Tris pH 7.9, 1 mM EDTA, 5% glycerol) and loaded onto a HiTrap Q column. RNA polymerase was eluted in a gradient from 100 mM to 1 M NaCl. RNA polymerase eluting at ~500 mM NaCl was pooled, concentrated, and made to 50% glycerol for −20 °C storage. *E. coli* RNA polymerase was purchased from New England Biolabs and σ70 was purified as described previously[16]. The *B. subtilis sigA* gene, cloned in pET-28a, was used to transform T7 Express *E. coli* (New England Biolabs). Cells were grown in LB to exponential phase at 37 °C with shaking before protein expression was induced for 3 h with 1 mM IPTG. Overexpressed σ$^A$ forms inclusion bodies, which were isolated by as outlined by Borukhov and Goldfarb[67]. Inclusion bodies were solubilised in denaturing His-Trap buffer (20 mM Tris pH 7.9, 5% glycerol, 600 mM NaCl, 8 M urea) and cell debris removed by centrifugation at 45,000 × *g* for 20 min. Solubilised inclusion bodies were loaded on a His-Trap column (GE Healthcare), which was washed with denaturing His-Trap buffer then stepwise increasing concentrations of imidazole. Fractions containing σ$^A$ were pooled and diluted with an equal volume of denaturing buffer (50 mM Tris pH 7.9, 8 M urea, 10% glycerol, 10 mM MgCl$_2$, 10 μM ZnCl$_2$, 1 mM EDTA, 10 mM DTT) before dialysis overnight in 2 l of reconstitution buffer (50 mM Tris pH 7.9, 200 mM NaCl, 20% glycerol, 10 mM MgCl$_2$, 10 μM ZnCl$_2$, 1 mM EDTA, 1 mM DTT). Refolded σ$^A$ was diluted 4 times in HiTrap buffer and loaded onto a HiTrap Q column (GE Healthcare). σ$^A$ was eluted in HiTrap buffer with a gradient of 50 mM NaCl to 1 M NaCl. Fractions containing σ$^A$ were pooled, concentrated, and made to 50% glycerol for −20 °C storage.

The *rok* gene was cloned in pET-21a and used to transform *E. coli* T7 Express (New England Biolabs). Cells were grown in LB to exponential phase at 37 °C with shaking before protein expression was induced for 3 h with 1 mM IPTG. Cell pellets were resuspended in His-Trap buffer (20 mM Tris pH 7.9, 5% glycerol, 600 mM NaCl) and lysed by sonication. Lysate was clarified by centrifugation at 45,000 × *g* for 20 min at 4 °C followed by filtration through a 0.45 μM PES filter. Lysate was applied to a His-Trap column which was then washed with His-Trap buffer followed by stepwise increases in imidazole concentration. Protein eluted by 200 mM imidazole was diluted three times in 20 mM Tris HCl pH 7.9, 5% (*v/v*) glycerol and applied to a Heparin column (GE Healthcare). Protein was eluted with step wise increasing concentrations of NaCl ranging from 150 mM NaCl to 1 M NaCl. The fractions containing Rok were pooled, concentrated, and made to 50% glycerol for −20 °C storage. *E. coli* H-NS was purified as described previously[16].

**in vitro transcription assays**. Each in vitro transcription reaction contained 0.01 μM DNA template, 0.05 μM *B. subtilis* RNAP/0.5 units *E. coli* core RNAP (New England Biolabs), 0.25 μM σ$^A$/σ70 in Transcription buffer (20 mM Tris pH 7.9, 40 mM KCl, and 10 mM MgCl$_2$). Reactions were started by addition of NTP mix to give final NTP concentrations of 200 μM ATP/GTP/CTP, 10 μM UTP and 2 μCi [α-32P] UTP. After 10 min incubation at 37 °C, reactions were stopped by the addition of an equal volume of formamide containing stop buffer. DNA templates consisted of the promoter and/or gene of interest cloned in plasmid pSR. Plasmid templates were isolated from *E. coli* using Qiagen Maxiprep kits. Reactions were resolved on an 8% (*w/v*) denaturing polyacrylamide gel, exposed on a Bio-Rad phosphor screen then visualised on a Bio-Rad Personal Molecular Imager using Quantity One software (version 4.6.9).

**Potassium permanganate footprinting**. KMnO$_4$ footprints were done using standard procedures with the following alterations[16]. DNA fragments were made by polymerase chain reaction amplification of pVEG cloned in plasmid pSR and labelled with 50 μCi [γ-32P] ATP using T4 Polynucleotide Kinase (New England

Biolabs). Reactions contained 10 nM labelled DNA, 100 nM RNAP, 500 nM $\sigma^{70}$/$\sigma^A$, 10 mM HEPES pH 8, 20 mM $CH_3COOK$, 0.1 mM DTT, 10 mM $Mg(C_2H_3O_2)_2$ and 100 µg ml$^{-1}$ BSA.

**Reporting summary**. Further information on research design is available in the Nature Research Reporting Summary linked to this article.

## Data availability

The raw RNA-seq and cappable-seq data generated in this study have been deposited in ArrayExpress under accession code E-MTAB-10777. Reference genomes NC000964.3 or U00096.3 (https://www.ncbi.nlm.nih.gov/nuccore/545778205) were used as appropriate. Results generated from processing of the sequencing data (e.g. to determine changes in gene expression or TSS signals) are available as Source Data or Supplementary Data files. The ChIP-seq data used were obtained from the ArrayExpress or NCBI databases for *E. coli* H-NS (E-MTAB-332) or *B. subtilis* Rok (PRJNA272948) respectively. All Original gel images are in Supplementary Fig. 7 and results obtained from quantification of band intensities are provided as Source Data. Source data are provided with this paper.

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

## Acknowledgements

This work was funded by a Wellcome Trust Investigator award (no. 212193/Z/18/Z) and a Leverhulme Trust project grant (no. RPG-2018-198) to D.C.G. We thank Joseph Wade, Steve Busby, and John Helmann for critical reading of the manuscript prior to submission. We also thank James Haycocks for proofreading the final version of the text.

## Author contributions

D.F., E.A.W. and D.C.G. designed the project. All experimental work was done by D.F. with support and advice from E.A.W. Bioinformatics was done by D.F. and D.C.G. A.E.M. and R.T.D. provided materials. D.F. and D.C.G. wrote the paper with all authors making edits as appropriate.

## Competing interests

The authors declare no competing interests.
