## [Peer Review File · Nature Communications]

Reviewers' Comments:

Reviewer #1:

Remarks to the Author:

This paper asks a fundamental question about how bacteria can avoid spurious transcription at AT-rich sequences in their genomes, including the large proportion of such sequences that are horizontally acquired.

In this work the authors compare AT-rich silencing proteins and the RNAP polymerases of *E. coli* (H-NS is the silencer) and *B. subtilis* (Rok is the silencer). Through a variety of approaches both global (e.g. cappable-seq mapping of 5' ends) and reductionist (footprinting and in vitro transcription) they provide compelling evidence that *B. subtilis* avoids spurious intragenic transcription because its RNAP (and associated sigma factors) are very specific whereas *E. coli*'s tends to be more promiscuous...able to start transcription from several AT rich sequences that are not functional as start sites in *B. subtilis*. Hence, *E. coli* needs to polymerize H-NS along AT-rich DNA whereas *B. subtilis* generally needs less silencing protein and it's generally concentrated at TSS sites and not within genes.

My biggest comments are in presentation:

1. *B. subtilis* is a rare firmicute (AT-rich Gm+) that has a mid-GC% genome and a silencer like Rok. In fact I think some strains of *Bacillus* don't have Rok...are they even more AT-rich. This is in contrast to H-NS, which is fairly universal in enterobacteria and vibrios. What about bacteria that are related to *B. subtilis* but don't have Rok? What does their RNAP look like? What is the distribution of foreign DNA or AT-rich islands in those species?

All that to say - this story could be re-framed to how AT-rich bacteria in general avoid promiscuous transcription. Has Rok changed how *B. subtilis* does basic core transcription (made the RNAP more or less promiscuous)? Or was Rok-mediated repression a recent adaptation in *B. subtilis* that was simply layered on top of an already stringent RNAP (because ancestors were AT-rich firmicutes)?

2. Rok has a slightly different sequence specificity than H-NS or other silencers. It's not just the TA steps, right? Please be more clear on the differences in sequence specificity that explain why Rok rarely polymerizes into the gene sequence. I think I understand why the lack of Rok in some AT-rich patches isn't a bad thing (because its RNAP is stringent)...but it doesn't explain why Rok doesn't polymerize into gene sequences. I think it would be interesting to see if what was defined by NMR studies lines up well with the Rok immunoprecipitation data to clarify this part of Rok function.

3. I'm a bit surprised that sigmaD from *E. coli* was functional with *B. subtilis* RNAP when expressed. The authors should state more clear prior evidence that this approach isn't prone to artifactual garbage. I wouldn't believe this result alone but it adds to and agrees with a lot of the other evidence they have for their model.

4. The mutations of the 2 arginines in the *B. subtilis* sigma were not well explained. The rationale for these mutations could be better explained up front.

5. The final figure is not particularly informative as drawn. It should show *B. subtilis* rejecting the spurious promoters.

Reviewer #2:

Remarks to the Author:

The manuscript entitled "Xenogeneic silencing strategies in bacteria are dictated by RNA polymerase promiscuity" by Forrest et al. describes the investigation of possible mechanisms of xenogeneic silencing in both *E. coli* and *B. subtilis*. The authors used TSS and NA-seq as well as publicly available datasets to argue that it is contrasting transcriptional promiscuity that is driving differences in xenogeneic silencing strategy in *B. subtilis* and *E. coli*. While the work is interesting, a number of major points need to be addressed before publication.

Major comments :

The data that are used to derive some of the conclusions do not support those conclusions (see underlined text) : "We demonstrate fundamentally different xenogeneic silencing strategies in *E. coli* and *B. subtilis* (Figure 6). In the former, RNA polymerase promiscuity creates a need for widespread transcriptional silencing by H-NS. Hence, xenogeneic silencing inhibits both mRNA expression and spurious intragenic transcription initiation. Conversely, Rok binds mostly to non-coding DNA and better RNA polymerase specificity reduces intragenic transcription."
You do not have these data to support such conclusions.

The +rpoD strain is induced using IPTG. It is unclear whether the control strain is also induced by IPTG. If not, TSS induced by IPTG (and not resulting from the expression of +rpoD) may confound the identification of sigma 70 TSS.

figure 4C : the TSS signal looks more 'noisy' in the +rpoD strain notably on the minus strand which would make sense. In the analysis described in figure XX, It would be interesting to also include the noise in addition to the TSS (TSS, I am assuming, has to pass a certain cutoff to be annotated as TSS) - this may better describe the +rpoD strain phenotype.

Correct explanations are missing for the majority of the analysis :

The bioinformatic part of the materials and methods is not explanatory enough : how are the sequence logos being done - was a background being considered ? for TSS : "For each strand, coverage was used to call TSSs at positions where read depth increased threefold or more, compared to the previous base, in both replicates". Is this meaning that a TSS is called if 3 reads are found overlapping ? Are the reads normalized in any way ?

For the part where external programs have been used : can the version of the program be clearly displayed and all the options specified (if different from default).

Generally, with this level of explanation, it is impossible to repeat what has been done.

Data access and missing information

Reviewers have no access to the raw data (The resource located at /arrayexpress/experiments/EMTAB-10777/ may have been removed, had its name changed, or has restricted access.). It is unclear the depth of sequencing (I did not find this information in the text and raw data is unavailable). Figures showing example of genomic positions have the label (example Figure 1) "Scales are identical for data obtained using wild type and Δ hns cells for each type of experiment" : Does this mean that the sequencing depth has been normalized or that the sequencing has been downsampled to the same number of reads ?

Minor comments :

[1] Figure 1a and d and Figure 2 a and c : it would be nice to have the sliding window with the AT/GC ratio plotted as an additional plot. Since H-NS is preferentially binding to AT rich regions, the AT/GC ratio profile should presumably follow the ChIP-seq profile of H-NS.

[2] Figure 1 b and e : Please have the two graphs on the same scales for comparison. The green points are masking the grey points : could they be on distinct plots.

[3] AT-rich DNA / AT-rich island / AT-rich sections : Are those all the same definitions ? How do you define an AT-rich island/sections/DNA (for example figure 4b) ?? I did not find this information in the material and method.

[4] Lines 145 - 160 : the many references to figure 1 should be figure 2 instead.

[5] "The result of the in vitro transcription assay is shown below the Figure 1b schematic. As expected, a 696 nt transcript was detected (lane 1) and subject to repression by Rok (lanes 2 and 3).. " This is Figure 2b not 1b. Also it should be noted that lane 3 exhibits a general inhibition of

transcription and not just for the 696 nt transcript. Same applies for Figure 2d.

[6] Supplementary Figure S1 : Can the authors label the axis? Why is the % of maximum binding signal adding to more than 100% ?

[7] "B. subtilis transcription start sites dependent on E. coli σ 70 have promoters that more closely resemble those identified in E. coli." I guess this is based on a visual inspection of the motif logos - If that's the case, the differences between all these logos are minor and rather suggests than confirms the fact that B. subtilis transcription start sites dependent on E. coli σ 70 have promoters that more closely resemble those identified in E. coli. Unless the author can demonstrate that the differences are significant, the title of figure S3 and associated text should reflect this.

Reviewer #3:

Remarks to the Author:

The manuscript describes and explains intriguing differences of the strategies bacteria use to silence promiscuous transcription of AT-rich DNA. In particular, the authors show that promoter selectivity of RNA polymerase associated with the housekeeping sigma factor is significantly higher for RNA polymerase-SigmaA in B.subtilis than for RNA polymerase-Sigma70 in E.coli. Therefore in B.subtilis RNA polymerase is less prone to initiate transcription at AT-rich sequences, while in E.coli H-NS is required to prevent transcription initiation by RNAP-Sigma70 at spurious promoters. The manuscript is of broad significance; it is concise in every respect.

Minor comments, suggestions for editorial modifications:

- 1) L. 229-230, Figure S4. When complementing E.coli delta-hns with Rok the protein level of Rok is likely to be crucial. Was the level determined? Was sufficient Rok present for silencing of AT-rich DNA in E.coli? How does the level compare to Rok levels in B.subtilis and to H-NS levels in E.coli?
- 2) In vitro transcription assays with E.coli RNAP-sigma70 revealed the full length transcript and shorter transcripts (Fig 4d and 5e). What is the argument that these shorter transcripts carry different 5'ends (spurious intragenic promoters) rather than different 3' ends (premature termination). Maybe the result obtained with Sigma70 versus Sigma70mut (Fig 5e) could be used to argue for spurious intragenic promoters?
- 3) Reading the manuscript the rationale for the choice of the promoters and genes used e.g. in in vitro assays is not always clear. The rationale is stated for comK and agaB (l. 164-167), while the basis for the choice of, yddK, ynaB, gmuB (used in Figs 4 and 5), as well as goxA and sunA (Figure 4d and 5e only) is less clear. Also, why was the veg promoter used for studying the discriminator and sigma specificity (Fig 3 and 5 and supplements)?
- 4) It was helpful if the source of the ChIP-seq data could also be specified in the results section.
- 5) L. 259 Re-phrase to "In B.subtilis σ A residue D222 corresponding to sigma70 R486 is negatively charged"?
- 6) L. 117 typo, cappable-seq

We thank the reviewers for their constructive comments. We have responded below and altered the manuscript text and figures where appropriate.

Reviewer #1 (Remarks to the Author):

This paper asks a fundamental question about how bacteria can avoid spurious transcription at AT-rich sequences in their genomes, including the large proportion of such sequences that are horizontally acquired.

In this work the authors compare AT-rich silencing proteins and the RNAP polymerases of *E. coli* (H-NS is the silencer) and *B. subtilis* (Rok is the silencer). Through a variety of approaches both global (e.g. cappable-seq mapping of 5' ends) and reductionist (footprinting and in vitro transcription) they provide compelling evidence that *B. subtilis* avoids spurious intragenic transcription because its RNAP (and associated sigma factors) are very specific whereas *E. coli*'s tends to be more promiscuous...able to start transcription from several AT rich sequences that are not functional as start sites in *B. subtilis*. Hence, *E. coli* needs to polymerize H-NS along AT-rich DNA whereas *B. subtilis* generally needs less silencing protein and it's generally concentrated at TSS sites and not within genes.

My biggest comments are in presentation:

1. *B. subtilis* is a rare firmicute (AT-rich Gm+) that has a mid-GC% genome and a silencer like Rok. In fact I think some strains of *Bacillus* don't have Rok...are they even more AT-rich. This is in contrast to H-NS, which is fairly universal in enterobacteria and vibrios. What about bacteria that are related to *B. subtilis* but don't have Rok? What does their RNAP look like? What is the distribution of foreign DNA or AT-rich islands in those species?

The reviewer is correct that some strains of *Bacillus* don't have Rok*. In these cases, the RNA polymerase sigmaA factor hardly differs at all to that of *B. subtilis* (as expected, because the protein is highly conserved). Compared to *B. subtilis*, these organisms have a similar distribution of foreign AT-rich DNA. As the reviewer (we think) eludes to, it may well be that Rok is dispensable because the RNA polymerase is more specific in *Bacillus* species. Indeed, this would fit with the lack of fitness defect observed upon deletion of *rok*.

*of course, they may well have something that they use instead of Rok, for the same task, which we don't know about.

All that to say - this story could be re-framed to how AT-rich bacteria in general avoid promiscuous transcription. Has Rok changed how *B. subtilis* does basic core transcription (made the RNAP more or less promiscuous)? Or was Rok-mediated repression a recent adaptation in *B. subtilis* that was simply layered on top of an already stringent RNAP (because ancestors were AT-rich firmicutes)?

We agree that our story could have been framed in this way and it's interesting to think about such questions. That said, models for how evolution proceeded in the distant past are difficult to test experimentally. Hence, we would like to keep our current narrative that compares the xenogeneic silencing systems of *E. coli* and *B. subtilis*.

2. Rok has a slightly different sequence specificity than H-NS or other silencers. It's not just the TA steps, right? Please be more clear on the differences in sequence specificity...

We think it's fair to say that all of the reasonably well-characterised silencers (H-NS, MvaT, Lsr2 and Rok) have slightly different sequence specificity. The most up-to-date comparison is provided by a

recent review (PMID: 34003286). The paraphrase the review, in general terms, all of the silencers prefer to bind sequences that are rich in A or T bases. Like H-NS and MvaT, Rok binds more tightly to sequences with TpA steps compared to sequences with continuous A or T tracts. Indeed, when binding site motifs for the different silencers are compared (see Figure 2b of PMID: 34003286) the presence of a TpA step is much more prominent in Rok targets (being almost completely conserved) than targets of the other silencers. Compared to H-NS, MvaT and Lsr2, Rok is less sensitive to the presence of G or C bases within A/T rich tracts.

We have made some alterations to the text at the start of the section “High-resolution mapping of the Rok regulated transcriptome in *Bacillus subtilis*” to clarify the role of TpA steps vs A or T tracts.

...that explain why Rok rarely polymerizes into the gene sequence. I think I understand why the lack of Rok in some AT-rich patches isn't a bad thing (because its RNAP is stringent)...but it doesn't explain why Rok doesn't polymerize into gene sequences.

Currently, it is not known why Rok rarely polymerises into the gene sequence. Our view is that this is more likely a consequence of Rok being present at insufficient levels to coat all of this DNA. This is something we elude to in the introduction with the sentence “It is noteworthy that Rok, present at ~1,500 molecules per cell, is much less abundant than H-NS in *E. coli*”.

I think it would be interesting to see if what was defined by NMR studies lines up well with the Rok immunoprecipitation data to clarify this part of Rok function.

We aren't certain which study the reviewer is referring to. Most likely, they are referencing PMID: 30252102 that determines a solution structure of Rok by NMR and uses an oligonucleotide array to identify the highest affinity targets for Rok. As mentioned above, Rok preferentially binds short AT-rich sequences with almost absolute conservation of a TpA step (see the below DNA sequence logo taken from PMID: 30252102). The three highest affinity sequences for Rok found in the paper are 5'-AACTA-3', 5'-TACTA-3', 5'-ATATA-3'. Prior to submitting our manuscript, we searched for these sequences in regions bound by Rok in ChIP-seq experiments (this was a suggestion made by John Helmann, who kindly read an initial draft of the paper). There is certainly a positive correlation between the occurrence of these sequences and the ChIP-seq signal. However, it's difficult to draw a firm conclusion from this observation. This is because these sequences are expected to be enriched in regions bound by Rok simply because the DNA is more AT-rich. For example, we'd also expect to find this sequence more frequently in regions of the *E. coli* genome bound by H-NS, by virtue of their high AT-content. As such, we decided not to include this analysis in the paper.

3. I'm a bit surprised that sigmaD from *E. coli* was functional with *B. subtilis* RNAP when expressed. The authors should state more clear prior evidence that this approach isn't prone to artifactual garbage. I wouldn't believe this result alone but it adds to and agrees with a lot of the other evidence they have for their model.

We understand why the reviewer might have this opinion. Our take is that the functionality of *E. coli* sigmaD in *B. subtilis* isn't surprising. We'd like to make the following points to explain our logic.

1. *E. coli* sigma70 and *B. subtilis* sigma are 55 % identical and 83 % similar (excluding the non-conserved region). Moreover, of the 11 amino acids that mediate interaction with core RNAP, 10 are identical between sigma70 and sigmaA. The remaining residue is a methionine in sigma70 and a leucine in sigmaA (i.e. two amino acids with very similar properties). An

alignment of the two factors, with amino acids for core RNAP recognition highlighted, is shown in Figure S5.

2. *B. subtilis* encodes a total of 19 alternative sigma factors that readily function with the core RNAP (http://www.subtiwiki.uni-goettingen.de/wiki/index.php/Sigma_factors). *E. coli* sigma70 resembles *B. subtilis* sigmaA far more closely than sigmaA resembles many of these alternative *B. subtilis* factors. For example, *B. subtilis* sigmaE is only 29 % identical and 64 % similar to *B. subtilis* sigmaA. Furthermore sigmaE has only 3 of the 11 amino acids used by sigmaA to bind core RNAP. Of the 8 remaining amino acids, only 2 are similar between sigmaE and sigmaA. Most strikingly, sigmaE completely lacks a region containing 3 of the amino acids used by sigmaA to bind core RNAP.
3. It has already been shown that *E. coli* sigma70, and *B. subtilis* sigmaA, can be swapped with respect to each other's cognate core RNAP (PMID: 11029421).
4. More generally, there are many papers showing that "foreign" sigma factors can be expressed in a given bacterium to drive transcription using the host core RNAP (e.g. PMIDs: 29361130, 25944046, 28319371 and 25232540).

We have made two changes to the text. First, when the sigma factor swapping experiments are introduced, we now point out the existing evidence that this type of approach has been used in the past. Second, when Figure S5 is introduced, we point of the similarity between sigma70 and sigmaA with respect to amino acids for binding the core RNA polymerase.

4. The mutations of the 2 arginines in the *B. subtilis* sigma were not well explained. The rationale for these mutations could be better explained up front.

We have expanded to the text to explain that we examined differences between sigma70 and sigmaA in the context of *E. coli* core RNA polymerase holoenzyme structures. In particular, we focused on differences in the properties of surface exposed amino acid side chains that might influence the interaction of the sigma factor with either core RNA polymerase or the DNA. This identified only two candidate side chains and these were the ones we followed up on.

5. The final figure is not particularly informative as drawn. It should show *B. subtilis* rejecting the spurious promoters.

We have made changes to the figure to improve clarity and show spurious promoters being rejected.

Reviewer #2 (Remarks to the Author):

The manuscript entitled "Xenogeneic silencing strategies in bacteria are dictated by RNA 8 polymerase promiscuity" by Forrest et al. describes the investigation of possible mechanisms of xenogeneic silencing in both *E. coli* and *B. subtilis*. The authors used TSS and NA-seq as well as publicly available datasets to argue that it is contrasting transcriptional promiscuity that is driving differences in xenogeneic silencing strategy in *B. subtilis* and *E. coli*. While the work is interesting, a number of major points need to be addressed before publication.

Major comments :

The data that are used to derive some of the conclusions do not support those conclusions (see underlined text) :

We were not sure what the reviewer meant by “see underlined text”. We have assumed they mean the full excerpt from our paper that they have quoted. Apologies if we misunderstood. Below we have explained the evidence for our statements.

“We demonstrate fundamentally different xenogeneic silencing strategies in *E. coli* and *B. subtilis* (Figure 6). In the former, RNA polymerase promiscuity creates a need for widespread transcriptional silencing by H-NS. Hence, xenogeneic silencing inhibits both mRNA expression and spurious intragenic transcription initiation.

There is a great deal of evidence that RNAP promiscuity creates a need for widespread silencing by H-NS in *E. coli*. This evidence is provided in the current paper and in prior studies from our lab. In 2014, we showed that deletion of *hns* in *E. coli* results in the appearance of many new transcription start sites that are predominantly inside genes (PMID: 24449106). At the time, we suggested that this was because AT-rich genes, bound by H-NS, contain many promoter-like sequences. In 2015 we showed computationally that promoter -10 elements are amongst the best binding targets for H-NS (PMID: 25638302). A subsequent paper in 2017 confirmed this experimentally and we showed that point mutations in -10 elements, present within H-NS bound genes, prevented the promiscuous behaviour of RNA polymerase (PMID: 28067866). Most recently, in 2020, we have shown that AT-rich DNA in general supports transcription by *E. coli* RNA polymerase by supporting non-specific interactions between the enzyme and DNA backbone (PMID: 32297955). The current paper also demonstrates the promiscuous behaviour of *E. coli* RNA polymerase, both *in vivo* (Figures 1a, 1b, 1c, 2c, 3a, 3b, and S4) and *in vitro* (Figures 2d, 4d, and 5e). Furthermore, we now narrow this promiscuous behaviour down to the sigma70 subunit of RNA polymerase and amino acid side chains R157 and R486.

We appreciate that the reviewer, and readers, may not be familiar with our prior work. At the same time, we can't go into the level of detail above in our current conclusions section. As a middle ground, we now back up the statements at the start of the conclusions section with the appropriate references and figure citations.

Conversely, Rok binds mostly to non-coding DNA...

This is shown clearly in Figure S1 (a summary of the complete ChIP-seq data) and for individual Rok bound genes elsewhere (Figure 2a, Figure 4c and Figure S6c).

and better RNA polymerase specificity reduces intragenic transcription.”

Our evidence for this is that *B. subtilis* RNA polymerase does not instigate spurious transcription (i.e. behave promiscuously) when provided with DNA templates that are targets for Rok. We observe a lack of promiscuity both *in vivo* when Rok is deleted (Figures 1d, 1e, 1f, 2a, and 4c) and *in vitro* on naked DNA templates (Figure 2b, 4d, and 5e). This contrasts markedly with the promiscuity of *E. coli* RNA polymerase in equivalent experiments presented throughout the paper.

You do not have these data to support such conclusions.

See above. We wonder if we were missing a more subtle point from the reviewer as there is plenty of support for the statements we made. Or perhaps we hadn't been clear enough.

The +rpoD strain is induced using IPTG. It is unclear whether the control strain is also induced by IPTG. If not, TSS induced by IPTG (and not resulting from the expression of +rpoD) may confound the

identification of sigma 70 TSS.

The control strain was not induced with IPTG. However, IPTG cannot be metabolised by *B. subtilis* and we know of no regulators that respond to this molecule in the organism used. As such, it's difficult to imagine how IPTG might lead to the appearance of new transcription start sites, downstream of promoters resembling the *E. coli* σ^{70} consensus, other than by inducing *rpoD* expression. We note that comparisons of the transcriptome, +/- IPTG induction of a given factor, are common in the *B. subtilis* literature and indicate no confounding effects of IPTG addition. For instance, IPTG induction of Spx expression identifies mRNAs subject to specific control by Spx/oxidative stress (PMID: 14597697) whilst IPTG induction of epitope tagged Hfq allows Hfq associated RNAs to be isolated (PMID: 23457461). Particularly noteworthy is the use of IPTG to control levels of RNase Y (PMID: 23326572). In these experiments, RNase Y induction alters global transcript abundance, but not the overall pattern of which genes are/aren't expressed. Examining Figure S1 of the latter paper reveals remarkably similar patterns of gene expression +/- IPTG.

Overall, if IPTG were regulating global transcription, we'd expect to see a common pattern of IPTG controlled genes emerge from prior studies, but this is not the case.

figure 4C : the TSS signal looks more 'noisy' in the +*rpoD* strain notably on the minus strand which would make sense. In the analysis described in figure XX, It would be interesting to also include the noise in addition to the TSS (TSS, I am assuming, has to pass a certain cutoff to be annotated as TSS) - this may better describe the +*rpoD* strain phenotype.

The reviewer is correct, although "noise" is arguably the wrong word. These are genuine TSSs giving lower level signals than the surrounding TSSs. We've picked out three such signals below, from Figure 4c, and have highlighted the TSS and corresponding -10 element in the sequence. As the reviewer points out, this further supports our argument. We can't be sure what the reviewer means by "figure XX" but we would guess they refer to Figure 4b. If so, then some (but not all) of these low level TSSs will be included as they passed our cut-off. However, missing some of the lower level signals is unavoidable.

3' - Bottom strand sequence in 3- to 5' direction with -10 elements and associated TSSs (from the +*rpoD* dataset) underlined and numbered.

```

ctattgagataacattaggagatagagtaataat taagtagaaggaattgaaaatttagcgggttacaattataagtaactaagttagttaaaaataaagtgataaattaaaagttagaataggccttttggaaatacaaaaa
aagacatactcaaccatttcattttctatcaattcattaaatatacaaaaattgggaggggataagccattt tgatatagtaatttgggacagaaaagcagagagaattatagaggggaattgcggcggtttcaataataggag
aaagagtgggaaattttaaagaaaagttatttaaaaaatatagtgaatagattgaaaaaaggtaaggagggtttaattttcaatttctcattagtagcattacattatgggtacaaatcctcottaattttctagatcgttcctt
atgaactaaaggtagttaaaggtgtaagtttgttat ttttgcaaaaataactctctttaaagtaaacagcatattttaaactat atcagaagggatcattttcctctacgtaagtagtagcaactgaaatggatgtggtaa
tcagagaagtagtagaagagagaaaagtggttcataataataatgaggatg tcgtaatttaaatatgagaaagtttaagaacccttaacgtttggttttctaaaaaatatttagtattgtagtgattgagaggcaagacgat
tagtataaaactaat tttcttggaaatggattacttatatttttcaattgctattttatgtcataggaagttataag gtttaaa taatattatggaactgagaatcgt-5'

```

Correct explanations are missing for the majority of the analysis :

The bioinformatic part of the materials and methods is not explanatory enough : how are the sequence logos being done - was a background being considered ?

The details of how sequence logos were made have now been added to the materials and methods. The logos we present did not make alterations for background but, when preparing the logos, we did try several different approaches to take background into account (e.g. see two adjacent logos).

The various background correction methods made very little difference to the sequence logos. Furthermore, it's not obvious which background correction is most appropriate (e.g. overall genome GC content, GC content of intergenic regions, GC content of only the identified promoter regions). Since all of these approaches to background selection have problems, and because they made little difference anyway, we opted for no background correction.

for TSS : "For each strand, coverage was used to call TSSs at positions where read depth increased threefold or more, compared to the previous base, in both replicates". Is this meaning that a TSS is called if 3 reads are found overlapping ?

No, as there will be many places where three reads will overlap but the read depth does not change from one base to the next. We mean exactly what we state, if the read depth at a given base is threefold higher than the prior base a TSS is called.

Are the reads normalized in any way ?

No, there is no need to do this to identify TSSs from cappable-seq data.

For the part where external programs have been used : can the version of the program be clearly displayed and all the options specified (if different from default).

These details have been added.

Generally, with this level of explanation, it is impossible to repeat what has been done.

We have addressed all of the above points by adding further details to the materials and methods.

Data access and missing information

Reviewers have no access to the raw data (The resource located at /arrayexpress/experiments/E-MTAB-10777/ may have been removed, had its name changed, or has restricted access.). It is unclear the depth of sequencing (I did not find this information in the text and raw data is unavailable).

The information is below. The reviewer can log in to ebi.ac.uk using the supplied details.

<http://www.ebi.ac.uk/arrayexpress/experiments/E-MTAB-10777>

Username: Reviewer_E-MTAB-10777

Password: ytoc7qdb

Figures showing example of genomic positions have the label (example Figure 1) “Scales are identical for data obtained using wild type and Δ hns cells for each type of experiment” : Does this mean that the sequencing depth has been normalized or that the sequencing has been downsampled to the same number of reads ?

We mean that the y-axis scale is the same in all of the panels. This detail has been added to the text. There is no downscaling or normalisation.

Minor comments :

[1] Figure 1a and d and Figure 2 a and c : it would be nice to have the sliding window with the AT/GC ratio plotted as an additional plot. Since H-NS is preferentially binding to AT rich regions, the AT/GC ratio profile should presumably follow the ChIP-seq profile of H-NS.

We agree that it would be nice to have this information as the H-NS signal does indeed closely track the AT-content of the DNA (this is evident in many prior publications). We didn't included the additional graph to avoid overcrowding the figure and so would prefer not to make this change.

[2] Figure 1 b and e : Please have the two graphs on the same scales for comparison. The green points are masking the grey points : could they be on distinct plots.

We understand why the reviewer suggests having the plots in 1b and 1e on the same scale. At the same time, doing this makes the data in 1e more compressed and so more difficult to assess. More important, we think, is that the scale for each of the two panels in 1b and 2e respectively are the same. We would prefer to keep the data as is and note that there is no reason to expect transcriptomic data for an *hns* knockout in *E. coli*, and *rok* knockout in *B. subtilis*, to naturally sit on exactly the same scales.

With respect to data point colour, and splitting the data across two graphs, we again would rather not make the change: the figure is already rather cramped. Note that there is only one data set in each plot and the data points are simply split into two colours (i.e. H-NS bound or not). Hence, unless there is another gene (or TSS) that has exactly the same response to deletion of *hns* (or *rok*) there is nothing “behind” any of the data points. Also, in those cases where the transcriptional response is identical, it could involve data points of the same colour.

Prior to submission, we did try formatting the figure according to the various suggestions from the reviewer and, in the end, thought the current version was clearest.

[3] AT-rich DNA / AT-rich island / AT-rich sections : Are those all the same definitions ? How do you define an AT-rich island/sections/DNA (for example figure 4b) ?? I did not find this information in the material and method.

By AT-rich, we mean DNA that has a higher AT-content than the genome average. We have added text to the introduction to make this clearer. The exact locations of AT-rich islands are provided in Table S4, which is referenced at the appropriate position of the materials and methods section.

[4] Lines 145 - 160 : the many references to figure 1 should be figure 2 instead.

This has been corrected.

[5] “The result of the in vitro transcription assay is shown below the Figure 1b schematic. As expected, a 696 nt transcript was detected (lane 1) and subject to repression by Rok (lanes 2 and 3).” This is Figure 2b not 1b. Also it should be noted that lane 3 exhibits a general inhibition of transcription and not just for the 696 nt transcript. Same applies for Figure 2d.

The figure number has been corrected. We have also added text to state transcription of the plasmid backbone, not just the 696 nt RNA, is reduced.

[6] Supplementary Figure S1 : Can the authors label the axis?

We do not know what the reviewer refers to here. In S1a the x- and y-axes are labelled “base pairs upstream (-) or downstream (+) of nearest start codon” and “% of maximum binding signal” respectively. In S1b the x- and y-axes are labelled “binding signal as % of maximum signal” and “% of bound DNA regions that are non-coding” respectively.

Why is the % of maximum binding signal adding to more than 100% ?

We assume the reviewer is referring to Figure S1a. None of the signals are more than 100% of the maximum. Also, the signals in the different regions combined are not expected to add up to 100%. These bars simply show what the signal is, at a given position, compared to the maximum signal. For analogy, consider plotting the average speed of a car across different sections of a journey. If the highest average section speed was 100 mph this would be the maximum signal. If a section of the journey was done at 50 mph this would be 50% of the maximum. In fact, all of the other sections could be done at 50% of the maximum speed and so would not add up to 100%.

[7] “*B. subtilis* transcription start sites dependent on *E. coli* σ 70 have promoters that more closely resemble those identified in *E. coli*.” I guess this is based on a visual inspection of the motif logos - If that’s the case, the differences between all these logos are minor and rather suggests than confirms the fact that *B. subtilis* transcription start sites dependent on *E. coli* σ 70 have promoters that more closely resemble those identified in *E. coli*. Unless the author can demonstrate that the differences are significant, the title of figure S3 and associated text should reflect this.

We do not claim statistical significance in the title of the figure legend (or anywhere else). Neither do we use the word “confirms”. Instead, we are careful to use qualitative language in the figure legend (i.e. promoters that more closely resemble those identified in *E. coli*). In the main body of the text we state how the motifs differ “sigma70 dependent promoters found in *B. subtilis* had misaligned - 10 elements and less AT-rich discriminators (Figure S3a). Conversely, canonical *B. subtilis* promoters did not exhibit these sequence features (Figure S3b,c)”. These are factually accurate statements.

Reviewer #3 (Remarks to the Author):

The manuscript describes and explains intriguing differences of the strategies bacteria use to silence promiscuous transcription of AT-rich DNA. In particular, the authors show that promoter selectivity of RNA polymerase associated with the housekeeping sigma factor is significantly higher for RNA polymerase-SigmaA in *B. subtilis* than for RNA polymerase-Sigma70 in *E. coli*. Therefore in *B. subtilis* RNA polymerase is less prone to initiate transcription at AT-rich sequences, while in *E. coli* H-NS is required to prevent transcription initiation by RNAP-Sigma70 at spurious promoters. The manuscript

is of broad significance; it is concise in every respect.

Minor comments, suggestions for editorial modifications:

1) L. 229-230, Figure S4. When complementing *E. coli* delta-*hns* with *Rok* the protein level of *Rok* is likely to be crucial. Was the level determined? Was sufficient *Rok* present for silencing of AT-rich DNA in *E. coli*?

We believe the reviewer is wondering if we have expressed *Rok* at sufficient levels in *E. coli* to permit potential silencing. Our experimental set up is designed to ensure this is the case. In the *E. coli* experiments, *Rok* is expressed from the *hns* promoter and so should be at levels at least equivalent to those of H-NS. In fact, as the *Phns::rok* fusion is plasmid encoded, the copy number will be elevated. Hence, *Rok* levels are likely higher than usual H-NS levels.

How does the level compare to *Rok* levels in *B. subtilis* and to H-NS levels in *E. coli*?

With respect to *Rok* levels in *B. subtilis*, these are known to be lower than H-NS levels in *E. coli* (see details in the intro).

2) In vitro transcription assays with *E. coli* RNAP-sigma70 revealed the full length transcript and shorter transcripts (Fig 4d and 5e). What is the argument that these shorter transcripts carry different 5' ends (spurious intragenic promoters) rather than different 3' ends (premature termination). Maybe the result obtained with Sigma70 versus Sigma70mut (Fig 5e) could be used to argue for spurious intragenic promoters?

As the reviewer notes, the observation that these shorter transcripts disappear with Sigma70mut suggests they result from intragenic promoters (i.e., differ at their 5' ends). In our prior work (PMID: 28067866), we showed point mutations in the promoter-like sequences of AT-rich genes prevents the synthesis of such short transcripts. In the same paper, we also showed that short sequences from within such genes can be cloned and have promoter activity.

3) Reading the manuscript the rational for the choice of the promoters and genes used e.g. in in vitro assays is not always clear. The rational is stated for *comK* and *agaB* (l. 164-167), while the basis for the choice of, *yddK*, *ynaB*, *gmuB* (used in Figs 4 and 5), as well as *goxA* and *sunA* (Figure 4d and 5e only) is less clear.

We chose these genes because they are *Rok* targets and not subject to promiscuous transcription by *B. subtilis* RNAP when *rok* is deleted. That said, this is a general feature of *Rok* targeted genes so, in that sense, the selection is rather arbitrary. We have add a little extra text to explain the selection as outlined to the reviewer.

Also, why was the *veg* promoter used for studying the discriminator and sigma specificity (Fig 3 and 5 and supplements)?

We chose the *veg* promoter because it is a widely used model promoter to study *B. subtilis* transcription (an equivalent for *E. coli* might be the *lacUV5* promoter). We'd tried to make this point but have slightly altered the text to make this clearer.

4) It was helpful if the source of the ChIP-seq data could also be specified in the results section.

We have moved the position of a citation in the relevant H-NS section to make this more obvious. Additionally, we have added a little extra text to the section “High-resolution mapping of the Rok regulated transcriptome in *Bacillus subtilis*”.

5) L. 259 Re-phrase to “In *B. subtilis* σ A residue D222 corresponding to sigma70 R486 is negatively charged”?

The suggested change has been made.

6) L. 117 typo, cappable-seq

This has been corrected.

Reviewers' Comments:

Reviewer #2:

Remarks to the Author:

We were not sure what the reviewer meant by “see underlined text”. We have assumed they mean the full excerpt from our paper that they have quoted. Apologies if we misunderstood. Below we have explained the evidence for our statements.

“We demonstrate fundamentally different xenogeneic silencing strategies in *E. coli* and *B. subtilis* (Figure 6). In the former, RNA polymerase promiscuity creates a need for widespread transcriptional silencing by H-NS. Hence, xenogeneic silencing inhibits both mRNA expression and spurious intragenic transcription initiation.

The underlined text must have gone away with formatting (underlying text : “We demonstrate fundamentally different xenogeneic silencing strategies in *E. coli* and *B. subtilis*”. The data ‘indicates’ rather than ‘demonstrates’ fundamentally different xenogeneic silencing strategies in *E. coli* and *B. subtilis*.

for TSS : “For each strand, coverage was used to call TSSs at positions where read depth increased threefold or more, compared to the previous base, in both replicates”. Is this meaning that a TSS is called if 3 reads are found overlapping ?

No, as there will be many places where three reads will overlap but the read depth does not change from one base to the next. We mean exactly what we state, if the read depth at a given base is threefold higher than the prior base a TSS is called.

If the prior base has 1 read and the next base has 3 reads, the TSS will be called ?. Thus, TSS can be called with coverage as low as 3 reads. What happens when the prior base on the genome has 0 read and the next base has 1 or 2 reads ? the increase from one base to the next is infinity, thus, should be a TSS under your definition. Should a position with one read be called at TSS ?

Are the reads normalized in any way ?

No, there is no need to do this to identify TSSs from cappable-seq data.

Arguably yes. TSS can be normalized according to the total number of mapped reads to take into effect the total number of reads per experiment.

Reviewers have no access to the raw data (The resource located at </arrayexpress/experiments/E-MTAB-10777/> may have been removed, had its name changed, or has restricted access.). It is unclear the depth of sequencing (I did not find this information in the text and raw data is unavailable).

It is still unclear the depth of sequencing in the manuscript. This is important given that the authors have not normalized their data to the total number of reads.

[1] Figure 1a and d and Figure 2 a and c : it would be nice to have the sliding window with the AT/GC ratio plotted as an additional plot. Since H-NS is preferentially binding to AT rich regions, the AT/GC ratio profile should presumably follow the ChIP-seq profile of H-NS.

We agree that it would be nice to have this information as the H-NS signal does indeed closely track the AT-content of the DNA (this is evident in many prior publications). We didn't include the additional graph to avoid overcrowding the figure and so would prefer not to make this change.

??? Since Rok protein recognises AT-rich DNA, it is a highly relevant information to have notably in figure 2a and c. overcrowding is really not an issue here.

[2] Figure 1 b and e : Please have the two graphs on the same scales for comparison. The green points are masking the grey points : could they be on distinct plots.

We understand why the reviewer suggests having the plots in 1b and 1e on the same scale. At the same time, doing this makes the data in 1e more compressed and so more difficult to assess. More important, we think, is that the scale for each of the two panels in 1b and 2e respectively are the same. We would prefer to keep the data as is and note that there is no reason to expect transcriptomic data for an hns knockout in E. coli, and rok knockout in B. subtilis, to naturally sit on exactly the same scales.

With respect to data point colour, and splitting the data across two graphs, we again would rather not make the change: the figure is already rather cramped. Note that there is only one data set in each plot and the data points are simply split into two colours (i.e. H-NS bound or not). Hence, unless there is another gene (or TSS) that has exactly the same response to deletion of hns (or rok) there is nothing "behind" any of the data points. Also, in those cases where the transcriptional response is identical, it could involve data points of the same colour. Prior to submission, we did try formatting the figure according to the various suggestions from the reviewer and, in the end, thought the current version was clearest.

I'm okay with keeping the scales distinct and agree that it would compress the data - nonetheless I do not agree that there is nothing "behind" any of the data points. The point sizes are very large and I would be not surprised that the H-NS bound TSS points are masking the H-NS free TSS.

[7] "B. subtilis transcription start sites dependent on E. coli σ 70 have promoters that more closely resemble those identified in E. coli." I guess this is based on a visual inspection of the motif logos - If that's the case, the differences between all these logos are minor and rather suggests than confirms the fact that B. subtilis transcription start sites dependent on E. coli σ 70 have promoters that more closely resemble those identified in E. coli. Unless the author can demonstrate that the differences are significant, the title of figure S3 and associated text should reflect this.

We do not claim statistical significance in the title of the figure legend (or anywhere else). Neither do we use the word "confirms". Instead, we are careful to use qualitative language in the figure legend (i.e. promoters that more closely resemble those identified in E. coli). In the main body of the text we state how the motifs differ "sigma70 dependent promoters found in B. subtilis had misaligned -10 elements and less AT-rich discriminators (Figure S3a). Conversely, canonical B. subtilis promoters did not exhibit these sequence features (Figure S3b,c)". These are factually accurate statements.

No. Based on the answers of the authors I'm assuming that their conclusion is based on manual visualization. Nonetheless, the heights of a letter is based on a given background and -yes- background correction changes the motif logo and in general there are many problems drawing conclusions on motifs logos that are SO similar. There are easy ways to visualize differences : for example <http://kplogo.wi.mit.edu/> allows you to see differences between two sets of data (sigma70 dependent promoters versus all promoters). Many more ways to identify differences are available.

The underlined text must have gone away with formatting (underlying text : “We demonstrate fundamentally different xenogeneic silencing strategies in *E. coli* and *B. subtilis*”). The data ‘indicates’ rather than ‘demonstrates’ fundamentally different xenogeneic silencing strategies in *E. coli* and *B. subtilis*.

We have reworded the text to say that we “propose”, rather than “demonstrate”, fundamentally different xenogeneic silencing strategies in *E. coli* and *B. subtilis*.

If the prior base has 1 read and the next base has 3 reads, the TSS will be called? Thus, TSS can be called with coverage as low as 3 reads.

Such positions are not called. Actually, our wording in the methods section was not quite accurate (now corrected). TSS are called at positions where read depth increases more than 3-fold, not 3-fold or more. Hence, if the prior base had 1 read, the next base would need at least 4 reads. Note that we only include TSSs that are found in both biological replicates. This helps to accurately select TSSs with comparatively low coverage.

What happens when the prior base on the genome has 0 read and the next base has 1 or 2 reads? the increase from one base to the next is infinity, thus, should be a TSS under your definition. Should a position with one read be called at TSS?

No, such instances are not counted as TSSs in our analysis (as the reviewer points, calculating $1/0$ produces a nonsensical answer). We have added this detail to the text.

Arguably yes. TSS can be normalized according to the total number of mapped reads to take into effect the total number of reads per experiment.

I’m wondering if we misinterpreted the reviewer’s original query “Are the reads normalized in any way?”. In initial assessment, this was the last sentence of the reviewer’s paragraph asking about TSS selection from capable-seq data. We assumed the reviewer wanted to know if read depth normalisation was needed prior to calling TSSs (it isn’t, since normalisation wouldn’t change relative coverage from one base to the next). However, as the reviewer maybe alludes to above, normalisation is needed to identify differences between experiments; sequencing depth variation could mask real changes in gene expression. EdgeR, the software that we used for differential expression analysis, automatically adjusts for sequencing depth discrepancies when calculating fold-changes and *P* values. We have added this detail to the methods section.

It is still unclear the depth of sequencing in the manuscript. This is important given that the authors have not normalized their data to the total number of reads.

See above, the data were normalised where direct comparisons between datasets were made. Regardless, the total number of mapped reads (provided below, and easy to determine using the FASTQ files deposited in ArrayExpress) varied very little between experiments. For instance, the number of mapped reads for the two wild type *B. subtilis* capable-seq replicates was 9.00 million and 8.64 million. For the equivalent experiments following *rpoD* expression in *B. subtilis* the mapped 9.24 and 8.81 million reads for each replicate.

Organism	Strain	Experiment	Replicate	Number of mapped reads
Bacillus subtilis subsp. subtilis str. 168	trpC2	Cappable-seq	a	8,998,563
Bacillus subtilis subsp. subtilis str. 168	trpC2	Cappable-seq	b	8,640,828
Bacillus subtilis subsp. subtilis str. 168	trpC2	RNA-seq	a	9,748,199
Bacillus subtilis subsp. subtilis str. 168	trpC2	RNA-seq	b	10,915,117
Bacillus subtilis subsp. subtilis str. 168	trpC2 amyE::Phyperspank - rpoD (spec)	Cappable-seq	a	9,236,575
Bacillus subtilis subsp. subtilis str. 168	trpC2 amyE::Phyperspank - rpoD (spec)	Cappable-seq	b	8,813,110
Bacillus subtilis subsp. subtilis str. 168	trpC2 Δrok::kanR	Cappable-seq	a	8,367,699
Bacillus subtilis subsp. subtilis str. 168	trpC2 Δrok::kanR	Cappable-seq	b	8,100,382
Bacillus subtilis subsp. subtilis str. 168	trpC2 Δrok::kanR	RNA-seq	a	11,610,234
Bacillus subtilis subsp. subtilis str. 168	trpC2 Δrok::kanR	RNA-seq	b	11,324,335
Bacillus subtilis subsp. subtilis str. 168	trpC2 ΔrpoE::kanR	Cappable-seq	a	11,447,221
Bacillus subtilis subsp. subtilis str. 168	trpC2 ΔrpoE::kanR	Cappable-seq	b	11,257,855
Escherichia coli str. K-12 substr. MG1655	K-12 F-λ- ilvG-rfb-50 rph-1	Cappable-seq	a	11,139,381
Escherichia coli str. K-12 substr. MG1655	K-12 F-λ- ilvG-rfb-50 rph-1	Cappable-seq	b	11,009,490
Escherichia coli str. K-12 substr. MG1655	K-12 F-λ- ilvG-rfb-50 rph-1	RNA-seq	a	9,184,767
Escherichia coli str. K-12 substr. MG1655	K-12 F-λ- ilvG-rfb-50 rph-1	RNA-seq	b	10,078,731
Escherichia coli str. K-12 substr. MG1655	K-12 F-λ- ilvG-rfb-50 rph-1 Δhns::kanR	Cappable-seq	a	11,364,381
Escherichia coli str. K-12 substr. MG1655	K-12 F-λ- ilvG-rfb-50 rph-1 Δhns::kanR	Cappable-seq	b	9,838,021
Escherichia coli str. K-12 substr. MG1655	K-12 F-λ- ilvG-rfb-50 rph-1 Δhns::kanR	RNA-seq	a	10,786,551
Escherichia coli str. K-12 substr. MG1655	K-12 F-λ- ilvG-rfb-50 rph-1 Δhns::kanR	RNA-seq	b	9,879,149
Escherichia coli str. K-12 substr. MG1655	K-12 F-λ- ilvG-rfb-50 rph-1 Δhns::kanR puc19/Phns-rok	Cappable-seq	a	14,349,459
Escherichia coli str. K-12 substr. MG1655	K-12 F-λ- ilvG-rfb-50 rph-1 Δhns::kanR puc19/Phns-rok	Cappable-seq	b	10,658,509

??? Since Rok protein recognises AT-rich DNA, it is a highly relevant information to have notably in figure 2a and c. overcrowding is really not an issue here.

We agree that the relationship between Rok binding and AT content is import. In our prior response, we noted that such a direct visual comparison has already been published (E.g. see Figures 2 and 3 of PMID: 21085634). Additionally, we did not want to overcrowd our figures. Hence, we opted not to include the comparison. This is just a matter of personal preference. We agree that the overcrowding is less of an issue in Figure 2. However, including a trace of AT-content in this instance, but not elsewhere, would produce inconsistent data presentation between Figures. As a compromise, we have produced two new Figure panels (Figures S1a and S1b). These show the complete genome-wide DNA binding of H-NS and Rok, plotted alongside AT-content, with the various regions we have investigated marked on the plots.

I'm okay with keeping the scales distinct and agree that it would compress the data - nonetheless I do not agree that there is nothing "behind" any of the data points. The point sizes are very large and I would be not surprised that the H-NS bound TSS points are masking the H-NS free TSS.

The reviewers original request was that we split each dataset across two plots, making a total of 8 panels in Figures 1b and 1e. In our view, the plots would then be illegibly small and the figure very cramped. This is why we did not want to make this revision. However, we agree with the reviewer that we could make the data point sizes smaller (although not by too much; they become difficult to see). We have made this revision. As a final comment, these datasets are presented only to show a key difference between H-NS and Rok. Deletion of H-NS alters signals at TSSs and across whole genes. On the other hand, deletion of Rok changes only signals for whole genes. This result is clear regardless of how the datapoints are presented. The observation is also clear for the individual gene targets shown in Figures 1a, 1d, 2a and 2c.

No. Based on the answers of the authors I'm assuming that their conclusion is based on manual visualization. Nonetheless, the heights of a letter is based on a given background and -yes background correction changes the motif logo and in general there are many problems drawing conclusions on motifs logos that are SO similar. There are easy ways to visualize differences : for example <http://kplogo.wi.mit.edu/> allows you to see differences between two sets of data (sigma70 dependent promoters versus all promoters). Many more ways to identify differences are available.

With specific reference to the point “background correction changes the motif logo” we showed in our prior response that such changes to our motifs are extremely minor. In addition, no background correction approach is perfect. Hence, as explained previously, we decided not to use background correction in our analysis. We would like to come back to the reviewer’s original request on this issue. The reviewer stated that “I guess this is based on a visual inspection of the motif logos... Unless the author can demonstrate that the differences are significant, the title of figure S3 and associated text should reflect this”. As we explained in our prior response, we did not make any claims regarding significance. As the reviewer did not specify how they would like the Figure title/associated text to be changed we made no alterations. The reviewer is now suggesting that we use a different analysis entirely. Given that we fully responded to the original query, and that no issues were raised by other reviewers, this seems unnecessary.